# Low-Rank Graphon Learning for Networks

**Xinyuan Fan**[*]
Department of Statistics and Data Science
Tsinghua University
Beijing, China
fxy22@mails.tsinghua.edu.cn

**Feiyan Ma**[*]
Weiyang College
Tsinghua University
Beijing, China
mafy21@mails.tsinghua.edu.cn

**Chenlei Leng**[†]
Department of Applied Mathematics
Hong Kong Polytechnic University
Hong Kong, China
chenlei.leng@polyu.edu.hk

**Weichi Wu**[†]
Department of Statistics and Data Science
Tsinghua University
Beijing, China
wuweichi@mail.tsinghua.edu.cn

## Abstract

Graphons offer a powerful framework for modeling large-scale networks, yet estimation remains challenging. We propose a novel approach that leverages a low-rank additive representation, yielding both a low-rank connection probability matrix and a low-rank graphon–two goals rarely achieved jointly. Our method resolves identification issues and enables an efficient sequential algorithm based on subgraph counts and interpolation. We establish consistency and demonstrate strong empirical performance in terms of computational efficiency and estimation accuracy through simulations and data analysis.

## 1 Introduction

With advances in data collection, modeling network data has become increasingly important across domains such as brain networks [Maugis et al., 2020], co-authorship networks [Isfandyari-Moghaddam et al., 2023], and biological systems [Kamimoto et al., 2023]. A key challenge is understanding the generative mechanisms underlying these networks, which informs tasks like studying dynamics [Pensky, 2019], link prediction [Gao et al., 2016], and community detection [Jin et al., 2021].

A powerful framework for modeling networks is based on exchangeable graphs, where node permutations leave the edge distribution invariant. By the Aldous-Hoover theorem [Kallenberg et al., 2005], such graphs are characterized by a graphon, a symmetric measurable function. Graphons offer a unified perspective, supporting tasks such as asymptotic analysis of subgraph counts [Bickel et al., 2011] and graph equivalence testing [Maugis et al., 2020]. They also underpin widely used models, including the stochastic block model (SBM) [Holland et al., 1983], random dot product graphs (RDPG) [Young and Scheinerman, 2007], and latent space models [Hoff et al., 2002].

A graphon is a symmetric, measurable bivariate function and serves as the limit object for sequences of dense graphs [Lovász and Szegedy, 2006]. Without further assumptions, it cannot be directly estimated from a single network. However, its eigenvalue-eigenfunction decomposition offers a practical solution: by truncating to the leading components—analogous to principal component analysis—we obtain a low-rank approximation. The resulting connection probability matrix $P$, constructed by evaluating the graphon at observed nodes, inherits this low-rank structure.

---

[*]Equal contribution.
[†]Corresponding author.

39th Conference on Neural Information Processing Systems (NeurIPS 2025).

**Contributions.** We propose a novel low-rank approach to graphon modeling that simultaneously captures the low-rank structure of both the graphon and the connection probability matrix. While prior work has focused primarily on estimating the connection matrix, our method uniquely recovers both the graphon function and the matrix with the *same rank*, offering a unified framework that fills a gap in existing literature.

The method builds on the key observation that a rank-$r$ matrix can be decomposed into a sum of $r$ rank-1 components. By counting $O(r)$ carefully selected subgraphs, we efficiently extract these components and solve the resulting system to estimate the connection matrix $P$. The graphon $f$ is subsequently recovered by sorting and interpolation. Crucially, while sorting is a powerful tool for recovering one-dimensional latent structures as in our approach, it is generally ineffective when applied directly to $P$ estimated by other methods, which lack the rank-1 aligned structure necessary for accurate graphon recovery. This sequential design ensures that the estimated graphon inherits the correct rank of $P$, leading to a tuning-free, scalable method that performs robustly across a range of settings, including sparse networks, as validated in our experiments.

In addition to our methodological advances, we deliver several key theoretical contributions that deepen the understanding of low-rank graphon models. First, we establish sharp perturbation bounds for solutions to stochastic systems of equations (Lemmas L.5 and L.6), shedding light on how estimation errors propagate through the model. Second, we prove a novel result (Lemma L.7) showing that, in low-rank graphon settings, the appropriately scaled number of fixed-length paths from a node uniformly approximates its conditional expectation given the latent variable. This insight forms the backbone of our eigenfunction estimation strategy and provides a fresh analytical tool for studying low-rank network models. Finally, all our convergence results (Theorems 3.1, 3.2, and 3.6) are derived under the sup-norm, delivering stronger *uniform* guarantees than those based on average-error metrics commonly used in the literature. Collectively, these contributions offer a robust theoretical foundation and significantly enhance the practical impact of our approach.

**Literature review.** Existing graphon estimation methods generally fall into two categories: those that directly estimate the graphon and those that estimate the connection probability matrix $P$. A central challenge in both is reconciling the low-rank structure of $P$ with that of the underlying graphon, often resulting in inconsistencies.

Graphon-based approaches include Olhede and Wolfe [2014], who approximate the graphon with a step function by partitioning it into blocks. Their method requires permutation maximization via a greedy algorithm, which is computationally intensive (see Section 4). Chan and Airoldi [2014] refine this by reordering nodes by degree and applying total variation minimization, assuming strictly monotonic marginals, an assumption that excludes models like the SBM. Moreover, the resulting $P$ is not guaranteed to be low-rank.

In contrast, $P$-based methods include Chatterjee [2015], who assume low rank and use Universal Singular-Value Thresholding (USVT), treating the adjacency matrix as a noisy version of $P$. Zhang et al. [2017] propose neighborhood smoothing to estimate $P$, achieving near-minimax optimality. Gao et al. [2016] introduce a combinatorial least-squares estimator, later extended by Wu et al. [2025] to non-exchangeable networks. However, these methods do not directly recover the graphon. The latent variables are unknown and unordered, preventing $P$ from being viewed as a lattice sample of the graphon, which complicates alignment. Moreover, further analysis of the estimated $P$, such as through eigenvalue decomposition, is challenging, especially to achieve sup-norm consistency. Most prior work focuses on mean squared error bounds. These limitations underscore the need for methods bridging graphon and $P$ estimation. Our approach offers a unified framework enabling sup-norm consistency in both.

Finally, we note that extensive research has been devoted to graphon models, therefore, we provide a broader literature review in Section B of the appendix.

**Structure.** The paper is organized as follows: Section 2 introduces the low-rank graphon framework. Section 3 presents the estimation procedures and algorithms for both $r = 1$ and $r \geq 2$. Section 4 reports simulation studies illustrating the method's efficiency and accuracy. Section 5 concludes with a discussion and future research directions. Additional details, including the significance of graphon estimation, time complexity, simulation settings, rank selection, sensitivity analysis, real data examples, proofs, and technical lemmas, are provided in the appendix.

**Notations.** For a real number $x$, $\lfloor x \rfloor$ denotes the greatest integer less than or equal to $x$. For positive real numbers $a$ and $b$, we define $a \vee b = \max(a, b)$ and $a \wedge b = \min(a, b)$. Let $\|A\|_F$ be the Frobenius norm of matrix $A$, and $A_{ij}$ denote its element at the $i$-th row and $j$-th column. For two sequences of positive real numbers $a_n$ and $b_n$, we write $a_n = O(b_n)$ or $a_n \lesssim b_n$ if there exist constants $N$ and $C$ such that $a_n \leq Cb_n$ for all $n > N$. For random variable sequences $X_n$ and $Y_n$, we write $X_n = O_p(Y_n)$ if for any $\varepsilon > 0$, there exists a constant $C_\varepsilon > 0$ such that $\sup_n \mathbb{P}(|X_n| \geq C_\varepsilon |Y_n|) < \varepsilon$.

## 2 Low-rank Approaches

We consider a random graph $\mathcal{G} = (V, E)$ within the graphon model framework. For $i = 1, \ldots, n$, where $n$ is the network size, each node $i$ is associated with an i.i.d. random variable $U_i \sim$ Uniform$(0, 1)$. The edges $E_{ij}$ are independently drawn as $E_{ij} \sim$ Bernoulli$(f(U_i, U_j))$ for $i < j$, where $f(\cdot, \cdot)$ is a symmetric, measurable function $f : [0, 1]^2 \to [0, 1]$, the graphon. We impose $E_{ii} = 0$ and $E_{ij} = E_{ji}$ for $i > j$. While we focus on undirected graphs without self-loops, our methods extend to graphs with self-loops or directed graphs. Many large-scale real-world networks exhibit low-rank features, such as group memberships or communities, which popular models like SBM and RDPG capture. For further discussions and real-data examples, see Athreya et al. [2018], Thibeault et al. [2024], and Fortunato [2010].

To incorporate low-rank structure into graphon models, we introduce the following parsimonious model:

$$f(U_i, U_j) = \sum_{k=1}^{r} \lambda_k G_k(U_i) G_k(U_j), \tag{1}$$

where $|\lambda_1| \geq |\lambda_2| \geq \cdots \geq |\lambda_r| > 0$, $G_k$ is a measurable function with $\int_0^1 G_k^2(u)\,du = 1$ for $k = 1, \ldots, r$, and $\int_0^1 G_k(u) G_l(u)\,du = 0$ for $k \neq l$. This model represents a truncated eigen-decomposition of the graphon, as suggested by the Hilbert-Schmidt theorem [Szegedy, 2011]. Model (1) includes both the SBM and RDPG as special cases. If the $G_k$ functions are step functions, it reduces to an SBM with $r$ blocks. When all $\lambda_k$ values are positive, the model simplifies to a rank-$r$ RDPG.

Introducing low-rank structures in graphon models enhances their ability to capture real-world network features, such as community structures and latent memberships, while also offering computational advantages through the models additive separability. We propose a novel, computationally efficient, and theoretically grounded method for estimating the connection probabilities $\{f(U_i, U_j)\}_{i,j=1}^n$ and the full graphon function $f$. Notably, practical methods for estimating general graphon functions in polynomial time are scarce [Gao and Ma, 2021].

## 3 Methodology and Theory

Let $p_{ij} = f(U_i, U_j)$ represent the connection probability between the $i$-th and $j$-th nodes, with $P = (p_{ij})_{i,j}$ as the connection probability matrix. To provide an intuitive understanding, we first focus on the estimation of $P$ and $f$ for $r = 1$ in Section 3.1. We then extend the discussion to the general case where $r \geq 2$ in Section 3.2.

### 3.1 $r = 1$: Low-rank Modeling with Rank-1

To provide clarity, we begin by considering the case where $r = 1$, i.e., $f(U_i, U_j) = \lambda_1 G_1(U_i) G_1(U_j)$. Without loss of generality, assume that $\inf_{u \in [0,1]} G_1(u) \geq 0$; otherwise, we can replace $G_1(u)$ with $|G_1(u)|$. A key observation is that the degree of the $i$-th node, denoted $d_i = \sum_j E_{ij}$, satisfies the following equation:

$$\frac{\mathbb{E}(d_i \mid U_i)}{n-1} = \frac{1}{n-1} \sum_{j \neq i} \int_0^1 f(U_i, U_j)\,dU_j = \lambda_1 G_1(U_i) \int_0^1 G_1(u)\,du, \tag{2}$$

which is proportional to $G_1(U_i)$. Furthermore, by Lemma L.1, we have the following bound:

$$\sup_{i=1,\dots,n} \frac{|d_i - \mathbb{E}(d_i \mid U_i)|}{n-1} = O_p\left(\sqrt{\frac{\log(n)}{n}}\right). \tag{3}$$

Thus, $G_1(U_i)$ can be learned by $\frac{d_i}{n-1}$, and as a result, $p_{ij}$ can be estimated by $\frac{d_i d_j}{(n-1)^2}$, up to a multiplicative constant. To account for the sparsity of the graph $\mathcal{G}$, we apply a moment estimation to determine this multiplicative factor. Note that the method of moments (MoM) is widely used in network analysis, for example, Bickel et al. [2011] employed MoM to estimate parameters based on subgraph counts, while Zhang and Xia [2022] analyzed the finite-sample distribution of MoM estimators for network motifs using Edgeworth expansions. The estimation procedure for $p_{ij}$ is summarized in Algorithm 1.

---

**Algorithm 1** Estimation for $\{p_{ij}\}_{i,j=1}^n$ in Rank-1 Model.

---

**Require:** The graph $\mathcal{G} = (V, E)$.
1: For $i = 1, \dots, n$, let $d_i = \sum_{j:j \neq i} E_{ij}$.
2: Let $c_1 = \sum_{i,j:i \neq j} E_{ij} / \sum_{i,j:i \neq j} d_i d_j$.
3: For any $(i, j)$ pair, $i \neq j$, let the estimator of $p_{ij}$ be $\hat{p}_{ij} = 1 \wedge (c_1 d_i d_j)$.
4: Let $\hat{p}_{ii} = 0$ for $i = 1, \dots, n$.
5: Output $\{\hat{p}_{ij}\}_{i,j=1}^n$.

---

Algorithm 1 is straightforward and leverages the low-rank assumption with $r = 1$. Its time complexity is $O(n^2)$, which is efficient given that there are $O(n^2)$ values of $p_{ij}$ to learn. In contrast, SVD-based methods (e.g., Xu [2018]) typically require $O(n^3)$ time complexity, making them less efficient. We remark that when the connection probability matrix is known a priori to be rank-1, a truncated version of Xu [2018] can, in principle, be accelerated to $O(n^2)$ time by computing only the leading singular value and its corresponding singular vectors through efficient procedures such as the power iteration method (see also Section J). It should be noted, however, that although the power iteration method is often effective in practice, it generally lacks rigorous convergence guarantees and theoretical justification. The run times for these methods are shown in Table 1 for various ranks $r$.

We now present the theoretical results for the estimates $\hat{p}_{ij}$.

**Theorem 3.1.** *For $r = 1$, assume that $\int_0^1 G_1(u) \, du > 0$. Applying Algorithm 1 to obtain the estimates $\hat{p}_{ij}$, we have*

$$\sup_{i,j} |\hat{p}_{ij} - p_{ij}| = O_p\left(\sqrt{\frac{\log(n)}{n}}\right).$$

The assumption in Theorem 3.1 is mild and does not require the continuity of the function $G_1$. This flexibility allows our model to accommodate various block structures, including the SBM with a rank-1 connection probability matrix. Additionally, the estimated connection probability matrix $\hat{P} = (\hat{p}_{ij})_{i,j}$ retains the rank-1 structure, consistent with the rank of $P$. The result $\sup_{i,j} |\hat{p}_{ij} - p_{ij}| = O_p\left(\sqrt{\frac{\log(n)}{n}}\right)$ also implies convergence in the Frobenius norm, specifically:

$$\frac{\|\hat{P} - P\|_F^2}{n^2} = O_p\left(\frac{\log(n)}{n}\right),$$

a standard metric used in the literature, such as in Zhang et al. [2017] and Gao et al. [2015].

Estimating the graphon function $f(u, v)$ is generally more challenging due to identification issues arising from measure-preserving transformations [Borgs et al., 2015, Diaconis and Janson, 2007, Olhede and Wolfe, 2014]. As a result, many prominent methods, including those in Gao et al. [2016] and Zhang et al. [2017], focus on estimating the connection probability matrix, as we do in Theorem 3.1. In the case with $r = 1$, we can mitigate the non-identifiability issue by defining a canonical, monotonically non-decreasing graphon through rearrangement. Specifically, let $G_1^\dagger(u) = \inf\{t : \mu(G_1 \leq t) \geq u\}$, where $\mu(\cdot)$ denotes the Lebesgue measure. As shown in Barbarino et al.

[2022], the function $G_1^\dagger(u)$ is the monotone rearrangement of $G_1(u)$, making it monotonically non-decreasing, left-continuous, and measure-preserving. Moreover, $G_1^\dagger(u)$ is continuous if $G_1(u)$ is continuous. Consequently, we can focus on the canonical graphon $f^\dagger(u,v) := \lambda_1 G_1^\dagger(u) G_1^\dagger(v)$.

To estimate $f^\dagger(u,v)$, we propose a degree sorting and interpolation method. Let $\sigma(k)$ denote the index $i$ corresponding to the $k$-th smallest value in the sequence $\{d_i\}_{i=1}^n$, i.e., $d_{\sigma(1)} \le d_{\sigma(2)} \le \cdots \le d_{\sigma(n)}$. Then, for any $(u,v) \in [0,1]^2$, we define

$$\hat{f}^\dagger(u,v) := 1 \wedge (c_1 h(u) h(v)),$$

where $h(v)$ is defined as follows. Let $s = v(n+1)$ and $k = \lfloor s \rfloor$. Then, $h(v) = d_{\sigma(k)}(k+1-s) + d_{\sigma(k+1)}(s-k)$, for $k \in [0,n]$, with the convention that $d_{\sigma(0)} = d_{\sigma(1)}$ and $d_{\sigma(n+1)} = d_{\sigma(n)}$. We remark that the degrees with ties can be ordered in any sequence, and the resulting $h(v)$ will remain unchanged. This further ensures the uniqueness of $\hat{f}^+(u,v)$.

**Theorem 3.2.** *For $r = 1$, assume that $G_1(u)$ is Lipschitz continuous on the interval $[0,1]$, i.e., there exists a constant $M > 0$ such that for any $u_1, u_2 \in [0,1]$, $|G_1(u_1) - G_1(u_2)| \le M|u_1 - u_2|$. Then,*

$$\sup_{u,v \in [0,1]} |\hat{f}^\dagger(u,v) - f^\dagger(u,v)| \overset{a.s., L_2}{\longrightarrow} 0, \ and \ = O_p\left(\sqrt{\frac{\log(n)}{n}}\right).$$

The estimation rate of our method matches that of Chan and Airoldi [2014], with convergence in the **sup-norm**, which is stronger than the mean squared error (MSE)-based approaches of Xu [2018], Olhede and Wolfe [2014], and Chan and Airoldi [2014]. While MSE focuses on average error, the sup-norm ensures uniform convergence, controlling the error at every point. This distinction underscores the robustness and precision of our approach.

## 3.2 $r > 1$: Low-rank Modeling with Rank-$r$

For $r \ge 2$, the connection probability $p_{ij} = f(U_i, U_j)$ in (1) is additive. To estimate $p_{ij}$, it is necessary to recover each $\lambda_k$ and $G_k$ for $k = 1, \ldots, r$. A central difficulty lies in the fact that estimating $\lambda_k$ requires eliminating the dependence on the unknown components $G_1, \ldots, G_r$. A key conceptual insight is that this disentanglement can be achieved by leveraging subgraph count statistics. Subgraphs, or "motifs", are crucial both theoretically [Maugis et al., 2020, Bravo-Hermsdorff et al., 2023, Ribeiro et al., 2021] and practically [Milo et al., 2002, Dey et al., 2019, Yu et al., 2019]. Their expectations are expressed via the graphon function. Consider a cycle of length $a$ passing through node $i$, whose count in the sample can be written as

$$C_i^{(a)} = \sum_{\{i_1, \ldots, i_{a-1}\} \in I_{a-1}} E_{ii_1} E_{i_{a-1}i} \prod_{j=2}^{a-1} E_{i_{j-1}i_j} \text{ for } a \ge 3, \tag{4}$$

where $I_a = \{i_1, \cdots, i_a \text{ distinct}, i_k \neq i, 1 \le k \le a\}$. Its expectation is given by

$$\mathbb{E}(C_i^{(a)}) = \left[\prod_{j=1}^{a-1}(n-j)\right] \sum_{k=1}^r \lambda_k^a, \tag{5}$$

which is independent of all $G_k$. Moreover, under an appropriate normalization, $C_i^{(a)}$ concentrates around $\mathbb{E}(C_i^{(a)})$, as established in Lemma L.4. Consequently, by counting cycles of different lengths, we can recover all $\lambda_k$.

Similarly, consider a simple path of length $a$ that has node $i$ as an endpoint, whose count in the sample is

$$L_i^{(1)} = \sum_{i_1} E_{ii_1}, \quad L_i^{(a)} = \sum_{\{i_1, \ldots, i_a\} \in I_a} E_{ii_1} \prod_{j=2}^a E_{i_{j-1}i_j} \text{ for } a \ge 2. \tag{6}$$

Its expectations are given by

$$\mathbb{E}(L_i^{(a)}|U_i) = \left[\prod_{j=1}^{a}(n-j)\right]\sum_{k=1}^{r}\lambda_k^a G_k(U_i)\int_0^1 G_k(u)\,du, \tag{7}$$

$$\mathbb{E}(L_i^{(a)}) = \left[\prod_{j=1}^{a}(n-j)\right]\sum_{k=1}^{r}\lambda_k^a\left(\int_0^1 G_k(u)\,du\right)^2. $$

Moreover, by Lemma L.4 and Lemma L.7, $L_i^{(a)}$, $\mathbb{E}(L_i^{(a)}|U_i), \mathbb{E}(L_i^{(a)})$ are close under suitable normalizations. Therefore, after estimating $\lambda_k$, we can first use $L_i^{(a)}$ to obtain an estimate of $\int_0^1 G_k(u)du$, and then proceed to estimate $G_k(U_i)$. Substituting these estimates into equation (1) yields an estimator for $p_{ij}$. To illustrate the above calculations, we provide a toy example in Section C, while the rigorous proofs are given in our theoretical derivations in the appendix.

We summarize the procedure in Algorithm 2. Note that in Algorithm 2, we add a standardization step in Line 4. It typically enhances performance in finite samples, benefiting both dense and sparse graphon settings.

---

**Algorithm 2** Estimation for $\{p_{ij}\}_{i,j=1}^n$ in Rank-$r$ Model.

**Require:** The graph $\mathcal{G} = (V, E)$.
1: For $i = 1, \ldots, n$, compute $L_i^{(a)}, 1 \le a \le r$ and $C_i^{(a)}, 3 \le a \le r+2$ defined in (6) and (4).
2: Solve the system of equations

$$\begin{cases} y_k \ge 0, \text{ for } 1 \le k \le r, |\hat{\lambda}_1| > \cdots > |\hat{\lambda}_r|, \\ \sum_{k=1}^{r}\hat{\lambda}_k^a = \frac{1}{\prod_{j=0}^{a-1}(n-j)}\sum_{i=1}^{n}C_i^{(a)} \text{ for } 3 \le a \le r+2, \\ \sum_{k=1}^{r}\hat{\lambda}_k^a y_k^2 = \frac{1}{\prod_{j=0}^{a}(n-j)}\sum_{i=1}^{n}L_i^{(a)} \text{ for } 1 \le a \le r. \end{cases} \tag{8}$$

to obtain $(\hat{\lambda}_1, \cdots, \hat{\lambda}_r, y_1, \cdots, y_r)$.
3: For $i = 1, 2, \ldots, n$, compute the estimators $\hat{G}_1(U_i), \cdots, \hat{G}_r(U_i)$ from

$$\frac{1}{\prod_{j=1}^{a}(n-j)}L_i^{(a)} = \sum_{k=1}^{r}\hat{\lambda}_k^a y_k G_k(U_i) \text{ for } 1 \le a \le r. \tag{9}$$

4: Compute the standardized estimators $\tilde{G}_1(U_i), \cdots, \tilde{G}_r(U_i)$ from

$$\tilde{G}_k(U_i) = \hat{G}_k(U_i)/\sqrt{\sum_{i=1}^{n}\hat{G}_k^2(U_i)/n}. \tag{10}$$

5: For each pair $(i, j)$, where $i \ne j$, estimate $p_{ij}$ as $\hat{p}_{ij} = \left[1 \wedge \left(0 \vee (\sum_{k=1}^{r}\hat{\lambda}_k\tilde{G}_k(U_i)\tilde{G}_k(U_j))\right)\right]$. Set $\hat{p}_{ii} = 0$ for $i = 1, \ldots, n$.
6: Output $\{\hat{p}_{ij}\}_{i,j=1}^n$.

---

*Remark* 3.3. The primary computational complexity of Algorithm 2 arises from counting lines and cycles within the graph. Notably, counting paths that allow repeated nodes is considerably simpler than counting simple paths, as the former can be achieved via matrix multiplication with a complexity of $O(n^\omega)$, where $\omega = 2.373$ [Williams, 2012]. Motivated by this observation, we propose an algorithm (Algorithm 3) in Section D with matrix multiplication time complexity, while preserving all theoretical guarantees from Theorems 3.6 and 3.9.

*Remark* 3.4 (Comparison with the spectral method for estimating the connection probability matrix). Spectral methods, such as USVT [Chatterjee, 2015], estimate the connection probability matrix through eigenvalues and eigenvectors. However, our approach differs in several ways. First, our goal is to estimate the graphon function, not just the connection probability matrix, which leads to a distinct methodology based on subgraph counts and moment-based techniques. Additionally, our method achieves the minimax rate for mean squared error up to a logarithmic factor, without requiring smoothness assumptions, unlike spectral methods that assume piecewise constant or Hölder-class smoothness [Xu, 2018]. Finally, for sparse graphons, our method outperforms USVT, as shown in Table 2.

We impose the following mild conditions for the consistency of $\hat{p}_{ij}$.

**Assumption 3.5.** Assume that: (i) $|\lambda_1| > \cdots > |\lambda_r| > 0$, $\int_0^1 G_k^2(u)du = 1$, for $1 \le k \le r$, and $\int_0^1 G_i(u)G_j(u)du = 0$ for $1 \le i \ne j \le n$, (ii) $\int_0^1 G_k(u)du \ne 0$, for $1 \le k \le r$, (iii) there exists a constant $K > 0$ such that $\max_{1 \le k \le r} \sup_{u \in [0,1]} |G_k(u)| \le K$.

Assumption 3.5 (i) ensures the identifiability of the functions $G_k$, similar to the eigengap condition in the RDPG model [Lyzinski et al., 2014]. Condition (ii) guarantees a unique solution for the system of equations (8), akin to the requirement in Bickel et al. [2011]. Condition (iii) is mild and typically holds for most graphon functions. Notably, we do not require $G_k$'s to be piecewise smooth, which broadens the applicability of our model for estimating the connection probability matrix. The theoretical result for $\hat{p}_{ij}$ is presented next.

**Theorem 3.6.** *For $r \ge 2$, under Assumption 3.5, when $n$ is sufficiently large, there exists an open set $U \subset \mathbb{R}^{2r}$ containing the point $(\lambda_1, \cdots, \lambda_r, \int_0^1 G_1(u)\,du, \cdots, \int_0^1 G_r(u)\,du)$ such that, with probability $1$, the system of equations in (8) has a unique solution within this region. Moreover, for $\hat{\lambda}_k, 1 \le k \le r, \hat{p}_{ij}$, we have $\max_{1 \le k \le r} |\hat{\lambda}_k - \lambda_k| = O_p(n^{-1/2})$, and $\sup_{i,j} |\hat{p}_{ij} - p_{ij}| = O_p(\sqrt{\log(n)/n})$.*

Theorem 3.6 shows that $\|\hat{P} - P\|_F^2/n^2 = O_p(\log(n)/n)$, achieving a rate matching the minimax rate (up to a logarithmic factor) in Gao et al. [2015]. The rate does not involve $r$ since we treat it as fixed. We leave the case where $r$ grows with $n$ for future work.

Estimating graphon functions is more challenging for $r \ge 2$ due to the model's additive structure. To address this, we introduce the following assumptions for learning the graphon function in the $r \ge 2$ case:

**Assumption 3.7.** Assume that: (i) At least one $G_k$ is strictly monotonically increasing; and (ii) All $G_k$'s are Lipschitz continuous with constant $M$.

Assumption 3.7 aligns with the "canonical form" for graphon functions, where the monotone function serves as the reference marginal function. This is similar to the identification criterion in Chan and Airoldi [2014], which requires the graphon to be monotone after integrating out one argument. Note that Assumption 3.7 is not needed if the goal is to estimate only the connection probability matrix. We remark that Assumptions 3.5 and 3.7 together imply that the graphon function belongs to the 1-Hölder class. In contrast, Olhede and Wolfe [2014] assumed only $\alpha$-Hölder smoothness with $0 < \alpha \le 1$ when estimating the graphon via histogram methods. However, their approach requires the selection of a bandwidth parameter.

Under Assumption 3.7, we proceed without loss of generality by assuming $G_1$ is the reference marginal graphon. The recovery of $G_1$ relies on the following insight: for i.i.d. samples $U_1, \cdots, U_n$, the $j$-th order statistic $U_{(j)}$ is close to $j/(n+1)$ with high probability. Consequently, $G_1(U_{(j)})$ serves as an approximation to $G_1(j/(n+1))$. Moreover, $G_1(U_{(j)})$ is itself the $j$-th smallest element among $\{G_1(U_i)\}_{i=1}^n$, and each $\hat{G}_1(U_i)$ provides a good approximation to $G_1(U_i)$. Hence, $G_1(j/(n+1))$ can be effectively estimated. Since $n$ is large and $G_1$ is Lipschitz continuous, piecewise linear interpolation yields a consistent approximation of $G_1$. The estimation strategy for the other functions $G_k, k = 2, \cdots, r$ follows an analogous approach.

Formally, we first sort the estimated pairs $(\hat{G}_1(U_i), \hat{G}_2(U_i), \cdots, \hat{G}_r(U_i))$ according to the first coordinate. Let $\gamma$ be a one-to-one permutation such that

$$\hat{G}_1(U_{\gamma(1)}) \le \hat{G}_1(U_{\gamma(2)}) \le \cdots \le \hat{G}_1(U_{\gamma(n)}).$$

After sorting, we denote the reordered pairs as $(\hat{G}_1(U_{\gamma(i)}), \hat{G}_2(U_{\gamma(i)}), \cdots, \hat{G}_r(U_{\gamma(i)}))$. We then define the function

$$h_1(u) = \hat{G}_1(U_{\gamma(1)})I(u(n+1) < 1) + \hat{G}_1(U_{\gamma(n)})I(u(n+1) \ge n)$$

$$+ \sum_{k=1}^{n-1} (k + 1 - u(n+1))\,\hat{G}_1(U_{\gamma(k)}) + (u(n+1) - k)\,\hat{G}_1(U_{\gamma(k+1)}))I(\lfloor u(n+1) \rfloor = k)$$

as an estimate of the function $G_1$. For $G_k, k \geq 2$, recognizing that $G_k$ is a function of $G_1$, we define:

$$h_k(u) = \hat{G}_k(U_{\gamma(1)})I(h_1(u) < \hat{G}_1(U_{\gamma(1)})) + \hat{G}_k(U_{\gamma(n)})I(h_1(u) \geq \hat{G}_1(U_{\gamma(n)}))$$

$$+ \sum_{k=1}^{n-1} \left( \frac{\hat{G}_1(U_{\gamma(k+1)}) - h_1(u)}{\hat{G}_1(U_{\gamma(k+1)}) - \hat{G}_1(U_{\gamma(k)})} \hat{G}_k(U_{\gamma(k)}) + \frac{h_1(u) - \hat{G}_1(U_{\gamma(k)})}{\hat{G}_1(U_{\gamma(k+1)}) - \hat{G}_1(U_{\gamma(k)})} \hat{G}_k(U_{\gamma(k+1)}) \right)$$

$$I(\hat{G}_1(U_{\gamma(k)}) \leq h_1(u) < \hat{G}_1(U_{\gamma(k+1)})).$$

Finally, we define

$$\hat{f}(u,v) := 1 \wedge \left[ 0 \vee \left( \sum_{k=1}^{r} \hat{\lambda}_k h_k(u) h_k(v) \right) \right] \tag{11}$$

as an estimate of the graphon $f(u,v)$.

*Remark* 3.8. In practice, we can relax the previous approach by considering each $G_i$ $(i = 1, \ldots, n)$ as a potential reference function. For each $G_i$, we estimate the graphon $f$ as described above referencing $G_i$, then compare the expected motif (e.g., triangles, stars) densities from the estimated $\hat{f}$ with the observed motif densities in the sample. By evaluating this criterion for all $i$, we select the $\hat{f}$ that best matches the empirical motif distributions.

Theorem 3.9 presents the theoretical result for this estimation. Since its proof follows directly from the proof of Theorem 3.2, we omit the details.

**Theorem 3.9.** *For $r \geq 2$, under Assumptions 3.5 and 3.7, the estimated graphon given by (11) satisfies*

$$\sup_{u,v \in [0,1]} |\hat{f}(u,v) - f(u,v)| \xrightarrow{a.s., L_2} 0, and = O_p(\sqrt{\log(n)/n}).$$

Our result is based on the *sup-norm*, providing a stronger uniform convergence guarantee compared to pointwise or average error metrics. The achieved rate matches Chan and Airoldi [2014], demonstrating the optimality and robustness of our method.

*Remark* 3.10. When $r$ is unknown, we can estimate it using a ratio-based method. Details are provided in Appendix H.

## 4   Numerics

In this section, we evaluate our method's effectiveness through extensive simulations. We assess the accuracy of the learned connection probability matrix $P$ using three metrics: Mean Squared Error (MSE) as $\|\hat{P} - P\|_F^2/n^2$, maximum error $\max_{i \neq j} |\hat{p}_{ij} - p_{ij}|$, and time cost. The code is available at https://github.com/Chiyuru/Low-Rank-Graphon-Learning-for-Networks.

Networks are generated using seven graphons listed in Table 5 (Appendix) with $n = 2000$. We also consider sparse counterparts, where edge probabilities follow $E_{ij} \sim$ Bernoulli$(\rho_n f(U_i, U_j))$, with $\rho_n = n^{-1/2}$ controlling sparsity. We conduct 100 independent trials per configuration and report the average metrics.

For comparison, we include the following methods: **Universal Singular Value Thresholding (USVT):** [Chatterjee, 2015], **Sort-and-Smooth (SAS):** [Chan and Airoldi, 2014], **Network Histogram (Nethist):** [Olhede and Wolfe, 2014], **Neighborhood Smoothing (N.S.):** [Zhang et al., 2017], and **Power Iteration (P.I.):** [Stoer et al., 1980].

All methods are implemented using the R functions provided by the respective authors with default parameters. Experiments were conducted on an Apple M1 machine with 16GB RAM, macOS Sonoma, and R 4.2.1. For efficiency, we modified Algorithm 2 slightly, as described in Section F in the appendix.

We summarize the results in Table 1 (dense graphons) and Table 2 (sparse graphons). In the dense graphon cases, our method consistently matches or outperforms others in both MSE and maximum error, achieving the best results in the first and fourth settings. It also matches the computational

Table 1: Results for dense graphons across 100 independent trials.

| ID | Method | MSE ($\times 10^{-4}$) | Std. MSE ($\times 10^{-6}$) | Max. error ($\times 10^{-2}$) | Std. Max ($\times 10^{-3}$) | Run time (s) | ID | MSE ($\times 10^{-4}$) | Std. MSE ($\times 10^{-6}$) | Max. error ($\times 10^{-2}$) | Std. Max ($\times 10^{-3}$) | Run time (s) |
|---|---|---|---|---|---|---|---|---|---|---|---|---|
| 1 | Ours | **1.275** | 3.871 | **5.817** | 4.955 | 0.121 | 2 | 2.452 | 7.806 | 8.114 | 5.819 | 0.539 |
|  | N.S. | 7.853 | 5.175 | 16.749 | 9.849 | 115.076 |  | 12.033 | 9.750 | 17.617 | 8.275 | 115.757 |
|  | Nethist | 4.237 | 8.330 | 5.980 | 11.474 | 16.705 |  | 9.867 | 24.220 | 16.962 | 44.037 | 16.744 |
|  | USVT | 1.282 | 3.863 | 5.837 | 4.994 | 13.587 |  | 2.403 | 7.593 | **7.977** | 5.740 | 14.629 |
|  | SAS | 19.120 | 16.865 | 85.000 | 0.000 | 1.273 |  | 39.888 | 29.339 | 78.134 | 37.486 | 1.250 |
|  | P.I. | 1.280 | 3.860 | 5.837 | 4.994 | 0.304 |  | **2.400** | 7.590 | **7.977** | 5.740 | 0.274 |
| 3 | Ours | 1.973 | 6.794 | 10.163 | 8.537 | 0.259 | 4 | **1.826** | 6.172 | **12.898** | 12.559 | 0.627 |
|  | N.S. | 8.337 | 14.186 | 17.329 | 8.566 | 114.694 |  | 4.388 | 8.413 | 17.902 | 11.686 | 108.569 |
|  | Nethist | 7.942 | 27.962 | 17.094 | 10.368 | 20.288 |  | 3.928 | 16.982 | 18.649 | 15.296 | 22.829 |
|  | USVT | **1.919** | 6.530 | **9.395** | 7.146 | 13.758 |  | 7.617 | 17.079 | 13.637 | 13.036 | 10.064 |
|  | SAS | 26.987 | 77.248 | 94.849 | 20.701 | 1.241 |  | 18.641 | 90.264 | 97.064 | 24.053 | 1.422 |
|  | P.I. | 1.920 | 6.530 | **9.395** | 7.146 | 0.328 |  | 1.890 | 6.420 | 13.400 | 13.927 | 0.654 |
| 5 | Ours | 1.774 | 6.204 | 9.676 | 7.804 | 1.507 | 6 | 2.582 | 7.017 | **9.319** | 7.808 | 0.682 |
|  | N.S. | 7.101 | 14.996 | 17.473 | 9.156 | 122.995 |  | 7.527 | 6.960 | 18.312 | 11.119 | 115.813 |
|  | Nethist | 7.729 | 31.838 | 18.559 | 15.486 | 23.804 |  | 9.548 | 244.015 | 22.573 | 107.570 | 19.441 |
|  | USVT | **1.769** | 6.078 | **9.661** | 7.871 | 13.684 |  | 2.383 | 6.082 | 10.052 | 8.343 | 11.169 |
|  | SAS | 28.703 | 115.215 | 89.861 | 49.103 | 1.104 |  | 18.456 | 16.344 | 95.000 | 0.000 | 1.500 |
|  | P.I. | 4.680 | 7.730 | 29.196 | 56.981 | 0.736 |  | **2.380** | 6.080 | 10.052 | 8.344 | 0.682 |
| 7 | Ours | 3.768 | 9.091 | 12.721 | 12.233 | 1.316 |  |  |  |  |  |  |
|  | N.S. | 6.596 | 6.372 | 17.611 | 9.760 | 126.270 |  |  |  |  |  |  |
|  | Nethist | 41.224 | 1574.245 | 59.567 | 96.247 | 20.238 |  |  |  |  |  |  |
|  | USVT | 3.644 | 7.580 | **12.613** | 11.696 | 11.640 |  |  |  |  |  |  |
|  | SAS | 20.552 | 22.529 | 90.000 | 0.000 | 1.701 |  |  |  |  |  |  |
|  | P.I. | **3.640** | 7.580 | **12.613** | 11.696 | 1.005 |  |  |  |  |  |  |

Table 2: Results for sparse graphons characterized by $\rho_n = 1/\sqrt{n}$.

| ID | Method | MSE ($\times 10^{-4}$) | Std. MSE ($\times 10^{-6}$) | Max. error ($\times 10^{-2}$) | Std. Max ($\times 10^{-3}$) | ID | MSE ($\times 10^{-4}$) | Std. MSE ($\times 10^{-6}$) | Max. error ($\times 10^{-2}$) | Std. Max ($\times 10^{-3}$) |
|---|---|---|---|---|---|---|---|---|---|---|
| 1 | Ours | **0.036** | 0.127 | 1.784 | 2.548 | 2 | **0.115** | 0.401 | 2.560 | 3.367 |
|  | N.S. | 19.224 | 45.848 | 99.665 | 0.000 |  | 61.897 | 249.576 | 99.161 | 0.002 |
|  | Nethist | 0.116 | 0.460 | **1.115** | 1.594 |  | 0.391 | 1.536 | 2.112 | 2.926 |
|  | USVT | 0.051 | 0.077 | 0.335 | 0.000 |  | 0.352 | 2.524 | **1.790** | 0.017 |
|  | SAS | 0.153 | 1.990 | 99.665 | 0.000 |  | 0.946 | 5.704 | 99.160 | 0.013 |
|  | P.I. | 0.249 | 0.735 | 1.500 | 0.000 |  | 0.132 | 0.490 | 2.951 | 4.110 |
| 3 | Ours | **0.075** | 0.284 | 3.079 | 4.494 | 4 | **0.043** | 0.398 | 3.840 | 8.423 |
|  | N.S. | 40.084 | 119.792 | 99.968 | 0.093 |  | 14.573 | 43.039 | 99.973 | 0.047 |
|  | Nethist | 0.249 | 1.056 | 2.840 | 14.842 |  | 0.111 | 0.642 | 2.485 | 12.126 |
|  | USVT | 0.314 | 0.952 | **2.093** | 0.039 |  | 0.102 | 0.648 | **2.202** | 0.075 |
|  | SAS | 0.200 | 2.163 | 99.956 | 0.426 |  | 0.078 | 0.945 | 89.227 | 253.955 |
|  | P.I. | 0.099 | 0.445 | 4.255 | 6.704 |  | 0.115 | 2.704 | 33.130 | 158.566 |
| 5 | Ours | **0.071** | 0.330 | 2.910 | 4.576 | 6 | **0.105** | 0.536 | **2.571** | 0.909 |
|  | N.S. | 34.490 | 115.091 | 99.993 | 0.039 |  | 42.530 | 69.027 | 99.984 | 0.099 |
|  | Nethist | 0.229 | 0.948 | 3.001 | 20.765 |  | 0.408 | 1.372 | 4.326 | 3.571 |
|  | USVT | 0.294 | 0.971 | **1.906** | 0.023 |  | 0.224 | 1.160 | 4.734 | 3.877 |
|  | SAS | 0.118 | 0.907 | 75.367 | 325.895 |  | 0.390 | 1.396 | 89.351 | 3.252 |
|  | P.I. | 0.170 | 4.392 | 21.174 | 101.657 |  | 0.415 | 2.055 | 2.821 | 4.047 |
| 7 | Ours | **0.091** | 0.585 | **2.139** | 1.440 |  |  |  |  |  |
|  | N.S. | 29.434 | 103.240 | 99.858 | 1.310 |  |  |  |  |  |
|  | Nethist | 0.359 | 1.688 | 3.123 | 1.715 |  |  |  |  |  |
|  | USVT | 0.410 | 1.306 | 4.808 | 1.855 |  |  |  |  |  |
|  | SAS | 0.915 | 3.112 | 99.604 | 0.113 |  |  |  |  |  |
|  | P.I. | 0.259 | 0.924 | 2.813 | 3.071 |  |  |  |  |  |

speed of SAS and P.I., while significantly outperforming other methods in efficiency. Notably, our method achieves accuracy similar to USVT, which is nearly minimax optimal for MSE under certain conditions [Xu, 2018], but with much lower computational complexity. Our method requires no tuning parameters, enhancing robustness across settings. In contrast, the power iteration method, though effective in some cases, lacks convergence guarantees and theoretical support.

In sparse graphon cases, our method excels, consistently outperforming all other approaches in MSE. This is expected, as it directly incorporates the sparsity parameter $\rho_n$ (see equation (9)). Additionally, we conducted additional simulations to investigate the performance of all methods as sparsity varies from $n^{-1/2}$ to 1, taking graphon 4 and 5 as illustrative examples. The results are shown in Table 7

(Appendix). As sparsity increases, the estimation error for all methods increases. However, our method consistently outperforms others in the sparse regime.

For estimation of functions $G_k$, Figure 1 shows the fitted $G_1$ and $G_2$ using our method, with estimates closely matching the true values. Notably, $G_2$ is continuous but not monotonic. The comparison between the estimated graphon and the true graphon, measured by the maximum entrywise error (i.e., the sup-norm), is reported in Table 3. The results demonstrate that the sup-norm errors of the graphon estimates remain well controlled, which is in line with the theoretical guarantees established in Theorems 3.2 and 3.9.

Table 3: Sup-norms for graphon estimation across 100 independent trials.

| ID | Sup-norm ($\times 10^{-2}$) | Std. Dev. ($\times 10^{-3}$) |
|----|------|------|
| 3 | 12.053 | 10.738 |
| 4 | 15.539 | 12.037 |

We provide additional simulation studies in Appendix G and Appendix H, examining rank selection when the true rank is unknown, the impact of rank mis-specification, and the scalability of our method. Moreover, Appendix I demonstrates the practical utility of our approach through two real-world applications: the Primary School dataset and the U.S. Political Blogs dataset.

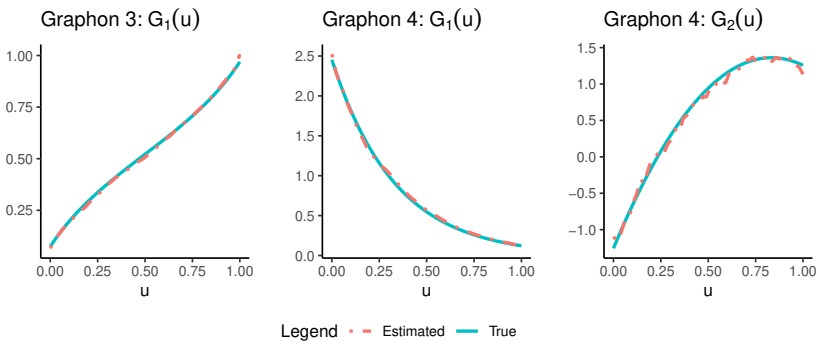

Figure 1: Estimation of $G_k$'s for the third and fourth settings.

## 5   Discussion

This work presents the first unified framework for simultaneously estimating both a low-rank connection probability matrix and its corresponding graphon. Traditional approaches treat these components separately, leading to inconsistencies and inefficiencies. By aligning their estimation, our method offers a more robust, consistent, and efficient solution for modeling network structures, particularly in low-rank graphon models. We propose a computationally efficient method based on subgraph counts, supported by rigorous theoretical guarantees for low-rank graphon models of fixed rank $r$. The method's effectiveness is validated through extensive simulations and real-data examples.

Future research directions include exploring convergence rates and optimality in sparse graphon models, optimizing subgraph selection for better efficiency and accuracy, and extending the method to cases where the rank $r$ increases with $n$, which could have significant implications for large-scale network analysis in high-dimensional or rapidly evolving networks.

## 6   Funding Disclosure

This work was supported by the High Performance Computing Center, Tsinghua University. Weichi Wu is supported by the NSFC No.12271287.

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

# A  Graphon Estimation Beyond the Connection Matrix

Graphon estimation provides a richer and more general description of network structure than the connection probability matrix. While the connection probability matrix captures marginal probabilities for edges in a specific network instance, it lacks information about how edges jointly interact. In contrast, the graphon $f$ defines a generative model over a family of networks and encodes global structural properties, including the distribution of motifs and other higher-order patterns.

For example, the expected density of a fixed motif $F$ (e.g., a triangle or star) in a large random graph generated from $f$ is given by its homomorphism density

$$\int_{[0,1]^{|V(F)|}} \prod_{(i,j)\in E(F)} f(x_i, x_j)\, dx_1 \cdots dx_{|V(F)|}.$$

For triangles, this quantity captures the limiting probability that three nodes form a triangle, a fundamentally joint property that the connection probability matrix alone cannot characterize. Similarly, the expected transitivity can be expressed as

$$\int_{[0,1]^3} f(x, y)f(y, z)f(z, x)\, dx\, dy\, dz \Big/ \int_{[0,1]^3} f(x, y)f(y, z)\, dx\, dy\, dz.$$

Graphon estimation thus enables direct prediction of such higher-order features. It also facilitates comparison between networks of different sizes by analyzing their underlying generative mechanisms, independently of the specific realizations captured in the connection probability matrix.

To substantiate the discussion, we conducted an experiment using graphons with IDs 4-6. We generated networks with 2000 nodes, randomly removed 10% of the nodes, estimated the graphon from the subgraph, and regenerated networks from the estimate. The mean absolute errors in triangle counts and transitivity (normalized appropriately) over 100 trials are shown below in Table 4.

Table 4: Triangle and transitivity errors across 100 repetitions.

| Graphon ID | Triangle Error ($\times 10^{-4}$) | | Transitivity Error ($\times 10^{-3}$) | |
|---|---|---|---|---|
| | Mean | Std. Dev. | Mean | Std. Dev. |
| 4 | 2.942 | 5.309 | 0.696 | 6.240 |
| 5 | 1.863 | 2.107 | 1.506 | 1.749 |
| 6 | 0.914 | 1.192 | 1.346 | 0.453 |

These consistently low errors demonstrate that key higher-order features are well preserved, even with subsampling, highlighting the structural fidelity and generalizability of graphon estimation.

# B  A Broader Literature Review

More recently, advances have been made in understanding the minimax rates and adaptivity of graphon estimation methods. Notably, [Klopp and Verzelen, 2019] rigorously analyzed optimal rates under the challenging cut distance, providing sharp risk bounds for several estimator classes. Oracle inequalities for network models, which allow for adaptivity in the presence of sparsity, were established by [Klopp et al., 2015], while [Donier-Meroz et al., 2023] extended the graphon framework to bipartite graphs with partial observability, further broadening the applicable scope of statistical models in network analysis. These works, along with related efforts by [Gao et al., 2015] and [Lei and Rinaldo, 2015], situate the current work within a rich landscape of research focused on both theoretical properties and practical algorithms for graphon estimation, highlighting the ongoing evolution of this field.

Matrix completion and matrix sensing are intimately related to graphon estimation, especially when considering networks with latent structure expressible through low-rank matrix representations. [Chen et al., 2014] addressed the general case of completing low-rank matrices, offering provable guarantees under mild assumptions, while the survey by [Candès and Recht, 2012] provides a foundational understanding of matrix completion via convex optimization techniques. These techniques inform approaches to graphon estimation, where the adjacency matrix of a network can often be approximated by a low-rank structure.

The study of smoothness assumptions further refines graphon estimation, particularly through the establishment of rates under Hölder or Sobolev smoothness conditions. For example, [Gao et al., 2015] analyzed community detection and graphon estimation under smoothness constraints, providing minimax rates and adaptive procedures. [Wu et al., 2025] introduced a flexible framework for network modeling beyond exchangeability, leveraging composite and Hölder-smooth graphon structures.

Finally, emerging works have sought to bridge graphon estimation with transfer learning and latent variable modeling. [Jalan et al., 2024] explored leveraging knowledge across related network models for improved inference, illustrating the potential for cross-network generalization. Collectively, these contributions provide a comprehensive landscape for graphon estimation, matrix completion, and smoothness-adaptive procedures, informing both the theoretical boundaries and practical approaches to latent structure recovery in large-scale networks.

The significance of statistical network analysis and graphon estimation continues to grow within the AI and machine learning communities, as evidenced by prominent work presented at leading conferences [Li et al., 2022a, Gaucher and Klopp, 2021, Araya Valdivia and Yohann, 2019]. Our methodology advances efficient and accurate network model estimation, with broad impact in domains such as social network analysis [Li et al., 2022b] and knowledge graphs [Nickel et al., 2015].

## C A Toy Example for Equations 5 and 7

Consider a rank-2 graphon $f(u, v) = \lambda_1 G_1(u) G_1(v) + \lambda_2 G_2(u) G_2(v)$. We illustrate the above result for $C_i^{(a)}$ using $a = 3$. Note that $n C_i^{(3)}/6$ corresponds to the number of triangles in the graph. Hence,

$$\mathbb{E}\left(\frac{n C_i^{(3)}/6}{n(n-1)(n-2)/6}\right) = \int_{[0,1]^3} f(u,v) f(v,w) f(w,u)\, du\, dv\, dw = \lambda_1^3 + \lambda_2^3,$$

where the last equality follows from the orthonormality of $\{G_j\}_{j=1,2}$. Moreover, we verify the conditional expectation of $L_i^{(3)}$. Since $L_i^{(3)} = \sum_{i_1, i_2 : i_1 \neq i_2,\, i_1 \neq i,\, i_2 \neq i} E_{ii_1} E_{i_1 i_2}$, we have

$$
\begin{aligned}
\mathbb{E}(L_i^{(3)} \mid U_i) &= \mathbb{E}\left[\sum_{i_1, i_2 : i_1 \neq i_2,\, i_1 \neq i,\, i_2 \neq i} \mathbb{E}(E_{ii_1} E_{i_1 i_2} \mid U_i, U_{i_1}, U_{i_2})\right] \\
&= (n-1)(n-2)\, \mathbb{E}[f(U_i, U_{i_1})\, f(U_{i_1}, U_{i_2}) \mid U_i] \\
&= (n-1)(n-2)\left(\lambda_1^2 G_1(U_i) \int_0^1 G_1(u)\, du + \lambda_2^2 G_2(U_i) \int_0^1 G_2(u)\, du\right).
\end{aligned}
$$

The derivation for more general cases proceeds analogously.

## D A Variant Algorithm and Time Complexity

In this section, we present a modified version of Algorithm 2 that preserves all theoretical guarantees from Theorems 3.6 and 3.9, while achieving the time complexity of matrix multiplication, $O(n^\omega), \omega = 2.373$. This variant algorithm is motivated by defining paths that permit node repetition. For $i = 1, \ldots, n$, define the lines and cycles *allowing repeated nodes* as:

$$\tilde{L}_i^{(1)} = \sum_{i_1} E_{ii_1}, \ \tilde{L}_i^{(a)} = \sum_{i_1, \cdots, i_a} E_{ii_1} \prod_{j=2}^{a} E_{i_{j-1} i_j} \text{ for } a \geq 2,$$

$$\tilde{C}_i^{(a)} = \sum_{i_1, \cdots, i_{a-1}} E_{ii_1} E_{i_{a-1} i} \prod_{j=2}^{a-1} E_{i_{j-1} i_j} \text{ for } a \geq 3.$$

These quantities can be computed efficiently. Specifically, let $E^a$ denote the $a$-th power of the adjacency matrix $E$. Then, we have:

$$\tilde{L}_i^{(a)} = \sum_{j \neq i} (E^a)_{ij}, \quad \tilde{C}_i^{(a)} = (E^a)_{ii}.$$

The variant algorithm (Algorithm 3) uses $\tilde{L}_i^{(a)}$ and $\tilde{C}_i^{(a)}$ instead of $L_i^{(a)}$ and $C_i^{(a)}$.

---

**Algorithm 3** Fast Estimation Procedure for $\{p_{ij}\}_{i,j=1}^n$ in Rank-$r$ Model.

---

**Require:** The graph $\mathcal{G} = (V, E)$.
1: For $i = 1, \ldots, n$, compute $\tilde{L}_i^{(a)} = \sum_{j \neq i}(E^a)_{ij}$ for $1 \leq a \leq r$, and $\tilde{C}_i^{(a)} = (E^a)_{ii}$ for $3 \leq a \leq r + 2$.
2: Set $L_i^{(a)} = \tilde{L}_i^{(a)}$ and $C_i^{(a)} = \tilde{C}_i^{(a)}$.
3: Follow from Line 2 of Algorithm 2 to learn $\{\hat{p}_{ij}\}_{i,j=1}^n$.
4: Output $\{\hat{p}_{ij}\}_{i,j=1}^n$.

---

*Remark* D.1 (Time Complexity of Algorithm 3). Since all $\tilde{L}_i^{(a)}$ and $\tilde{C}_i^{(a)}$ for $1 \leq i \leq n$ and $1 \leq a \leq r$ can be computed using matrix multiplication, which has a time complexity of $O(n^{2.373})$, the overall time complexity of Algorithm 3 is also $O(n^{2.373})$.

To analyze the theoretical properties, we present a key lemma showing that $\tilde{L}_i^{(a)}$ and $L_i^{(a)}$ (as well as $\tilde{C}_i^{(a)}$ and $C_i^{(a)}$) are sufficiently close, such that their differences do not impact the results of Theorem 3.6 and Theorem 3.9.

**Lemma D.2.** *For a rank-$r$ model, under the assumptions of Theorem 3.6, we have:*

$$\max_{1 \leq i \leq n} \max_{1 \leq a \leq r} \left| \frac{\tilde{L}_i^{(a)} - L_i^{(a)}}{\prod_{j=1}^a (n-j)} \right| = o_p\left(\frac{1}{\sqrt{n}}\right), \quad \max_{1 \leq i \leq n} \max_{3 \leq a \leq r+2} \left| \frac{\tilde{C}_i^{(a)} - C_i^{(a)}}{\prod_{j=1}^{a-1}(n-j)} \right| = o_p\left(\frac{1}{\sqrt{n}}\right).$$

With Lemma D.2 established, it follows straightforwardly that the following theorem holds.

**Theorem D.3.** *Theorem 3.6 and Theorem 3.9 remain valid when the fast estimation procedure described in Algorithm 3 is applied.*

# E   Graphons in Simulation Studies

We list the graphons in Table 5.

Table 5: List of graphons. We learn three rank-1 graphons using Algorithm 1 and four rank-$r \geq 2$ graphons using Algorithm 3.

| ID | Graphon $f(u,v)$ | Rank of $f(u,v)$ |
|----|------------------|------------------|
| 1 | $0.15$ | 1 |
| 2 | $\frac{1.5}{(1+\exp(-u^2))(1+\exp(-v^2))}$ | 1 |
| 3 | $\frac{1}{5}\left(\tan\left(\frac{\pi}{2}u\right) + \frac{7}{6}\right)\left(\tan\left(\frac{\pi}{2}v\right) + \frac{7}{6}\right)$ | 1 |
| 4 | $0.95\exp(-3u)\exp(-3v) + 0.04(3u^2 - 5u + 1)(3v^2 - 5v + 1)$ | 2 |
| 5 | $\frac{1}{2}(\sin u \sin v + uv)$ | 2 |
| 6 | $0.05 + 0.15I(u < 0.4, v < 0.4) + 0.25I(u > 0.4, v > 0.4)$ | 2 |
| 7 | $0.1 + 0.75I(u, v < \frac{1}{3}) + 0.15I(\frac{1}{3} < u, v \leq \frac{2}{3}) + 0.5I(u, v > \frac{2}{3})$ | 3 |

# F   A Remark on Computaiton

In Algorithm 3, we employ $\tilde{L}_i^{(a)}$ and $\tilde{C}_i^{(a)}$ as approximations for $L_i^{(a)}$ and $C_i^{(a)}$, enabling efficient computation. Though their equivalence has been proven in Theorem D.3, applying certain corrections in practice can improve finite-sample performance. Specifically, we define:

$$\check{L}_i^{(3)} = \tilde{L}_i^{(3)} - \tilde{L}_i^{(2)} - (\tilde{L}_i^{(1)})^2,$$
$$\check{C}_i^{(4)} = \tilde{C}_i^{(4)} - \tilde{L}_i^{(2)} - (\tilde{L}_i^{(1)})^2,$$
$$\check{C}_i^{(5)} = \tilde{C}_i^{(5)} - 2(\tilde{L}_i^{(1)} - 2)\tilde{C}_i^{(3)}$$
$$- \frac{1}{n}\left(\sum_{k=1}^n \tilde{L}_k^{(1)}\right)\tilde{C}_i^{(3)} - 2\sum_{k=1}^n E_{ik}\tilde{C}_k^{(3)},$$

and use $\check{L}_i^{(3)}$, $\check{C}_i^{(4)}$, and $\check{C}_i^{(5)}$ to replace $\tilde{L}_i^{(3)}$, $\tilde{C}_i^{(4)}$, and $\tilde{C}_i^{(5)}$, respectively, in Algorithm 3. These corrections ensure $\check{L}_i^{(3)} = L_i^{(3)}$, $\check{C}_i^{(4)} = C_i^{(4)}$, and $\check{C}_i^{(5)}$ is closer to $C_i^{(5)}$ compared to $\tilde{C}_i^{(5)}$.

# G   More Simulations

## G.1   Additional Simulation Results on Scalability

To show the scalability of our method, we vary $n$ from 1000 to 2000 for the rank-2 settings and report the results and time costs in Table 6. Our method consistently shows a decrease in both metrics as $n$ increases, which is consistent with our theoretical results.

Table 6: Results for rank-2 settings with varying node sizes between 1000 and 2000.

| ID | MSE ($\times 10^{-4}$) | MSE S.D. ($\times 10^{-4}$) | Max. Error ($\times 10^{-2}$) | Max. Error S.D. ($\times 10^{-2}$) | Node Size $n$ |
|----|-----|-----|-----|-----|-----|
| 4 | 3.677 | 0.143 | 16.618 | 1.745 | 1000 |
|   | 3.067 | 0.126 | 15.772 | 1.419 | 1200 |
|   | 2.457 | 0.091 | 14.507 | 1.490 | 1500 |
|   | 2.043 | 0.064 | 13.564 | 1.269 | 1800 |
|   | 1.826 | 0.062 | 12.898 | 1.256 | 2000 |
| 5 | 3.552 | 0.159 | 13.003 | 1.083 | 1000 |
|   | 2.995 | 0.396 | 12.252 | 1.819 | 1200 |
|   | 2.365 | 0.086 | 10.732 | 0.878 | 1500 |
|   | 2.001 | 0.311 | 10.444 | 2.244 | 1800 |
|   | 1.774 | 0.062 | 9.676 | 0.780 | 2000 |
| 6 | 5.576 | 0.185 | 12.119 | 1.160 | 1000 |
|   | 4.533 | 0.155 | 11.665 | 1.177 | 1200 |
|   | 3.535 | 0.108 | 10.378 | 0.938 | 1500 |
|   | 2.890 | 0.073 | 9.772 | 0.964 | 1800 |
|   | 2.582 | 0.070 | 9.319 | 0.781 | 2000 |

For time costs, we plot the logarithm of runtime against the logarithm of $n$ in Figure 2, with $n$ varying from 200 to 11000. The observed asymptotic trend exhibits linear growth, aligning with the theoretical computational complexity.

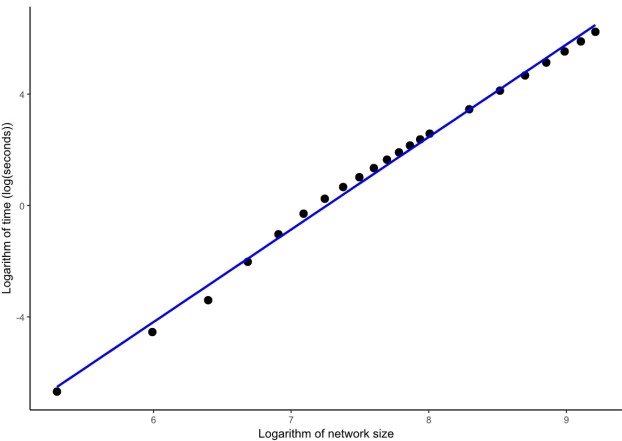

Figure 2: Average runtime of our algorithm as a function of node size, computed over 100 repetitions. The blue line indicates the best least-squares fit.

## G.2   Additional Simulation Results when Assumption 3.7 is Violated

Theoretically, Assumption 3.7 serves as a regularization condition for graphon estimation, and similar conditions are necessary and can be commonly found in the literature (e.g., Chan and Airoldi [2014]). If this assumption is violated, no theoretical guarantees for graphon estimation (Theorem 3.9) can be made; however, the estimation of the probability matrix (Theorem 3.6) still works

as expected. Practically, even if Assumption 3.7 is slightly violated, the estimated graphon function can still be reasonably close. We illustrate this with an example. Let $G_1(u) = \sqrt{\frac{6}{17}}(\sqrt{u} + 1)$, $G_2(u) = \sqrt{\frac{300}{17}}(\sqrt{x} - \frac{7}{10})$, which are non-Lipschitz orthogonal functions. We apply our method to estimate the graphon function, and the estimated $G_1$ and $G_2$ are plotted below. It is evident that the estimates exhibit the correct trend, and will be more accurate as the sample size increases.

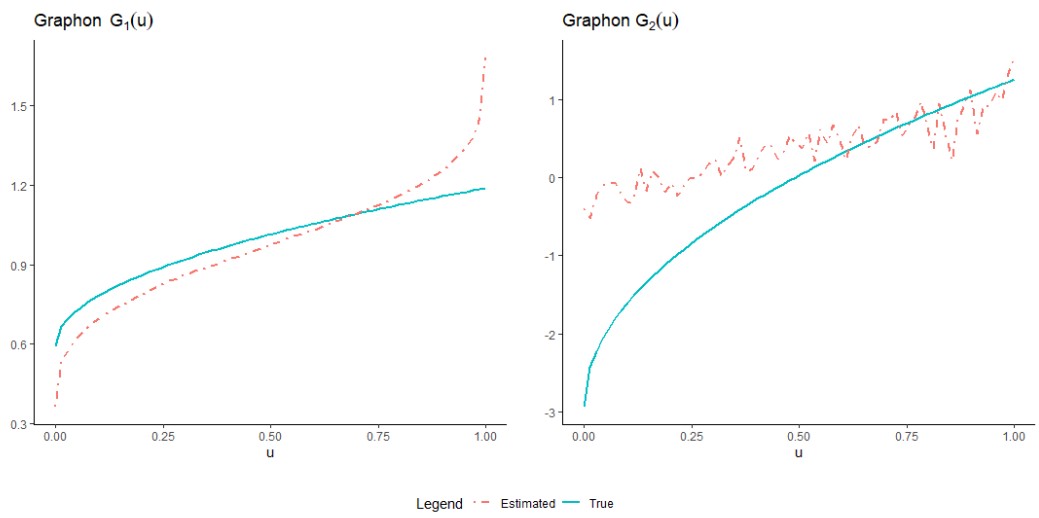

Figure 3: Estimation of the components of the graphon function for the proposed setting.

### G.3 Additional Simulation Results when the Sparsity Varies

Taking graphon 4 and 5 as illustrative examples, we conducted additional simulations to investigate the performance of all methods as sparsity varies from $n^{-1/2}$ to $1$, . The results are shown in Table 7.

## H  Selecting $r$ when it is unknown

In this section, we propose a method for selecting $r$ when its value is unknown. Since $\hat{\lambda}_k$ approximates $\lambda_k$ by Theorem 3.6, we can estimate $r$ incrementally, starting from $r = 1$. When $|\hat{\lambda}_k|$ is significantly larger than 0, but $|\hat{\lambda}_i|$ for $i \geq k + 1$ are close to 0, we select $r = k$. The detailed procedure for selecting $r$ is summarized in Algorithm 4.

---

**Algorithm 4** Procedure for selecting $r$.

---

**Require:** Graph $\mathcal{G} = (V, E)$, threshold $\tau$.
 1: For $i = 1, \ldots, n$, compute $\tilde{C}_i^{(3)}$. Set $k = 1$.
 2: For $i = 1, \ldots, n$, compute $\tilde{C}_i^{(k+3)}$.
 3: Solve the system of equations in (8) with $3 \leq a \leq k + 3$ and $r = k + 1$ to obtain $(\hat{\lambda}_1, \ldots, \hat{\lambda}_{k+1})$.
 4: **if** $\left| \frac{\hat{\lambda}_{k+1}}{\hat{\lambda}_k} \right| \leq \tau$, **then**
 5:     Choose $r = k$ and **return** $r$.
 6: **end if**
 7: Set $k = k + 1$ and go back to Line 2.
 8: **Output** $r$.

---

We remark that Line 4 in Algorithm 4 applies the eigenratio method, which is a well-established approach in the statistics literature (e.g., Lam and Yao [2012], Ahn and Horenstein [2013]). It offers

Table 7: Results for sparse graphons across 100 independent trials.

| Graphon ID | Sparsity Parameter $\rho_n$ | Method | MSE $(\times 10^{-4})$ | Std. dev of MSE $(\times 10^{-6})$ | Max. Error $(\times 10^{-2})$ | Std. dev of Max. Error $(\times 10^{-3})$ |
|---|---|---|---|---|---|---|
| | | Ours | 0.178 | 0.663 | 5.722 | 9.922 |
| | | N.S. | 31.285 | 140.98 | 99.907 | 0.15 |
| | $n^{-1/3}$ | USVT | 0.389 | 1.964 | 6.163 | 24.638 |
| | | Nethist | 1.193 | 7.809 | 7.815 | 0.278 |
| | | SAS | 0.597 | 3.787 | 99.72 | 0.985 |
| | | Ours | 0.752 | 3.501 | 10.022 | 12.695 |
| | | N.S. | 8.833 | 86.337 | 98.761 | 0.416 |
| 4 | $n^{-1/6}$ | USVT | 1.348 | 6.356 | 12.44 | 11.336 |
| | | Nethist | 13.66 | 89.165 | 27.699 | 1.149 |
| | | SAS | 5.356 | 32.228 | 99.597 | 2.321 |
| | | Ours | 1.116 | 4.874 | 12.412 | 14.103 |
| | | N.S. | 3.264 | 10.317 | 38.125 | 129.087 |
| | $n^{-1/10}$ | USVT | 2.161 | 10.221 | 15.706 | 14.102 |
| | | Nethist | 34.658 | 221.149 | 45.841 | 2.907 |
| | | SAS | 14.681 | 59.42 | 93.407 | 72.598 |
| | | Ours | 0.258 | 1.205 | 4.93 | 7.207 |
| | | N.S. | 29.992 | 170.322 | 99.969 | 0.167 |
| | $n^{-1/3}$ | USVT | 0.9 | 3.584 | 6.231 | 5.969 |
| | | Nethist | 3.316 | 10.137 | 6.766 | 0.074 |
| | | SAS | 0.821 | 4.783 | 97.805 | 101.723 |
| | | Ours | 1.145 | 34.613 | 11.757 | 40.912 |
| | | N.S. | 3.624 | 6.686 | 20.814 | 98.466 |
| 5 | $n^{-1/6}$ | USVT | 3.039 | 11.992 | 12.511 | 9.675 |
| | | Nethist | 30.673 | 72.197 | 23.993 | 0.395 |
| | | SAS | 9.165 | 18.182 | 74.31 | 132.569 |
| | | Ours | 1.338 | 33.349 | 10.409 | 30.081 |
| | | N.S. | 4.852 | 10.989 | 16.784 | 9.73 |
| | $n^{-1/10}$ | USVT | 4.626 | 18.243 | 15.865 | 13.431 |
| | | Nethist | 58.583 | 185.89 | 39.152 | 5.418 |
| | | SAS | 9.281 | 21.39 | 93.009 | 68.993 |

robustness to scaling and noise, especially in settings where the eigenvalues decay gradually or have non-uniform magnitudes (see also Cai et al. [2024]). In our context, the use of the ratio helps highlight significant drops in the eigenvalue sequence while mitigating the influence of slow decay or scale variability. Algorithm 4 selects the correct rank $r$ with high probability as the threshold $\tau$ asymptotically approaches zero at a certain rate.

We apply Algorithm 4 to select $r$ for the 3rd, 6th and 7th settings in Table 5, with $\tau = 0.2$. Theoretically, the threshold $\tau$ should asymptotically tend to 0. For finite sample simulations, we choose 0.2 as a heuristic value. The results are presented in Table 8. These results demonstrate that Algorithm 4 is effective in most cases.

In our simulations in the main paper, we assume the rank $r$ is known. Estimating $r$ first and then applying our method leads to minimal differences because $r$ can be chosen correctly with high probability by our method (see Table 8) in our scenarios. For example, we apply our rank selection algorithm to the 6th setting, and there is only an 8% probability of making an incorrect selection. The MSE when selecting $r$ first and then running our method is $2.4 \times 10^{-4}$ (with a standard error of $0.9 \times 10^{-6}$), which is close to the values reported in the main article.

Table 8: Selection of $r$ for the 3rd, 6th and 7th settings across 100 independent trials.

| ID | True $r$ | Estimated $r$ | | | |
|---|---|---|---|---|---|
| | | 1 | 2 | 3 | $\geq 4$ |
| 3 | 1 | 100 | 0 | 0 | 0 |
| 6 | 2 | 0 | 92 | 0 | 8 |
| 7 | 3 | 0 | 0 | 89 | 11 |

Moreover, we assess the impact of an incorrect rank selection by testing cases where $r = 1$ is mis-specified as $r = 2$. The results, presented in Table 9, show that the differences in performance are minimal compared to the original settings in Table 1, highlighting that our method is robust to over-estimation of the rank $r$.

Table 9: Results for rank-1 settings when their rank is mis-selected as 2.

| ID | MSE $(10^{-4})$ | Std. MSE $(10^{-4})$ | Max error $(10^{-2})$ | Std. Max $(10^{-2})$ |
|---|---|---|---|---|
| 1 | 1.45 | 0.475 | 6.274 | 1.226 |
| 2 | 6.11 | 6.650 | 18.186 | 18.17 |
| 3 | 1.92 | 0.067 | 9.399 | 0.693 |

For the case where $r = 2$ is mis-specified as $r = 1$, the corresponding results are summarized in Table 10. As shown in the table, the estimation error increases significantly when rank-2 graphons are incorrectly selected as rank 1, since an essential component is omitted during estimation. However, our rank selection experiments (see also Table 8) indicate that while the selected rank may sometimes be higher than the true rank, we seldom select a rank lower than the actual one, which would otherwise lead to large errors.

Table 10: Results summary with runtime statistics.

| ID | MSE $(\times 10^{-4})$ | Std. MSE $(\times 10^{-6})$ | Max. error $(\times 10^{-2})$ | Std. Max $(\times 10^{-3})$ |
|---|---|---|---|---|
| 4 | 15.559 | 5.939 | 30.818 | 17.708 |
| 5 | 1.810 | 0.620 | 10.320 | 8.962 |
| 6 | 71.987 | 29.280 | 15.086 | 2.447 |

# I Real-world Data Analysis

We applied our method to two real-world datasets to evaluate its performance and robustness in practical scenarios.

## I.1 Primary School Student Contact Data

Firstly, we applied our method to real contact data from a primary school, collected by the SocioPatterns project[3] using active RFID devices that recorded data every 20 seconds. On October 1st, 2009, from 8:40 to 17:18, contact data were gathered for 236 individuals, resulting in 60,623 records. We constructed an undirected graph where nodes are connected if individuals had at least one contact. Using Algorithm 4 with a threshold $\tau = 0.25$, we selected $r = 4$ based on Table 11. We subsequently learned the connection probability matrix with Algorithm 3. The resulting heatmap, shown on the left of Figure 4, aligns with expectations for real-world interactions.

Table 11: Learned eigenvalues for the contact data.

| Rank $r$ | $\hat{\lambda}_1$ | $\hat{\lambda}_2$ | $\hat{\lambda}_3$ | $\hat{\lambda}_4$ | $\hat{\lambda}_5$ |
|---|---|---|---|---|---|
| 2 | 0.264 | 0.159 | | | |
| 3 | 0.266 | 0.146 | 0.0593 | | |
| 4 | 0.271 | 0.118 | 0.118 | -0.0992 | |
| 5 | 0.272 | 0.117 | 0.0813 | -0.0423 | -0.00721 |

Assuming Assumption 3.7, we learn the graphon function, using $G_1$ as the reference marginal graphon. The learned functions $h_1, \ldots, h_4$ are shown on the right of Figure 4. The learned graphon function $\hat{f}(u, v)$ for any $(u, v) \in [0, 1]^2$ is computed using equation (11).

## I.2 U.S. Political Blogs Data

We evaluate our method on a larger real-world dataset. We consider the U.S. Political Blog Dataset [Adamic and Glance, 2005], which consists of 1490 nodes. This dataset captures the hyperlink net-

---
[3]http://www.sociopatterns.org

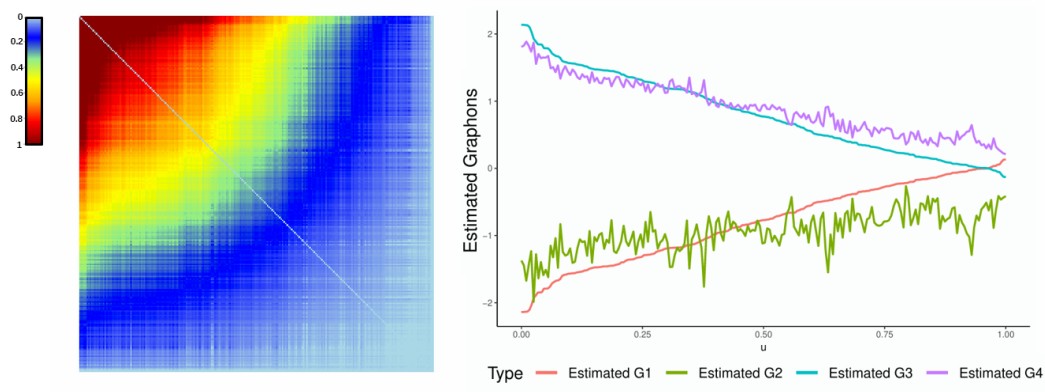

Figure 4: Learned $P$ (Left) and learned graphon (Right).

work among political blogs during the 2004 U.S. presidential election, where each node represents a blog labeled as either liberal or conservative, and approximately 19000 edges between these blogs.

Using Algorithm 4 with a threshold $\tau = 0.25$, we selected rank $r = 2$ based on Table 12.We mention that our choice of rank is consistent with the known structure of the network, in which blogs affiliated with the same political orientation tend to form two distinct communities, characterized by dense intra-community and sparse inter-community connections.

Table 12: Learned eigenvalues for the political blog data.

| Rank $r$ | $\hat{\lambda}_1$ | $\hat{\lambda}_2$ | $\hat{\lambda}_3$ |
|---|---|---|---|
| 2 | 0.0668 | 0.0326 | |
| 3 | 0.0692 | 0.0304 | 0.00649 |

We subsequently estimated the connection probability matrix using Algorithm 3. The resulting heatmap, shown in Figure 5, reveals a relatively sparse structure, reflecting the nature of the observed political blog network.

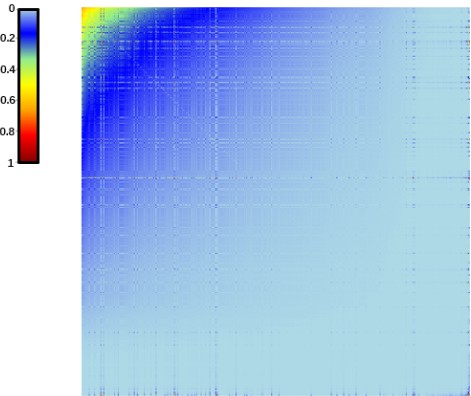

Figure 5: Learned connection probability matrix $P$ for U.S. political blog data.

Assuming Assumption 3.7 holds, we learn the graphon function using $G_1$ as the reference marginal graphon. The learned functions $h_1$ and $h_2$ are presented in Figure 6. The estimated graphon function $\hat{f}(u, v)$ for any $(u, v) \in [0, 1]^2$ can be computed according to equation (11). The learning procedure demonstrates promising performance when applied to large-scale real-world networks, and effectively detects the latent structure of low-rank networks.

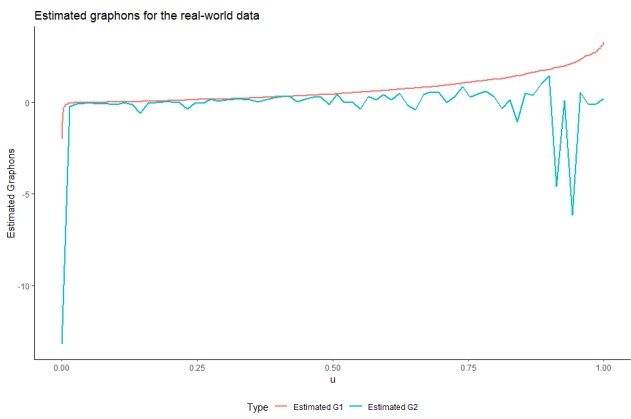

Figure 6: Estimated components of the graphon for U.S. political blog data.

## I.3 Validating the Quality of the Real-data Estimates

Using the U.S. political blog dataset as an illustrative example, we demonstrate the advantages of our method through extensive comparisons with existing approaches.

Specifically, we estimated the underlying graphon from the observed network and computed the expected densities of various motifs, such as triangles, squares, and 5-cycles. For competing methods, we generated synthetic networks from their estimated connection probability matrices and performed motif counting on these networks. We then compared the absolute differences in motif counts–normalized by the total possible in a complete graph of the same size–between the original and generated networks. The results, summarized in Table 13, show that our method consistently achieves smaller errors in motif prediction than competing approaches.

Table 13: Motif counting errors and runtime for each method.

| Method | Triangle counting error ($\times 10^{-4}$) | Runtime (s) | Square counting error ($\times 10^{-5}$) | Runtime (s) | 5-cycle counting error ($\times 10^{-6}$) | Runtime (s) |
|---|---|---|---|---|---|---|
| Ours | 0.252 | 0.167 | 0.761 | 0.198 | 3.848 | 0.175 |
| N.S. | 1.709 | 119.812 | 3.758 | 120.837 | 9.039 | 123.863 |
| Nethist | 8.087 | 19.082 | 1.809 | 20.523 | 4.383 | 22.427 |
| USVT | 4.449 | 15.971 | 1.712 | 17.098 | 5.149 | 19.439 |
| SAS | 1.327 | 3.018 | 2.256 | 4.730 | 4.960 | 5.237 |

Moreover, as shown in Table 13, our approach significantly improves computational efficiency. After estimating the graphon, the expected density of any motif in networks of any size can be quickly approximated using the plug-in method, with only a constant number of matrix operations per sample. In contrast, existing methods require explicit motif counting in generated networks, which is computationally expensive (see Jin et al. [2025] for details). This blend of efficiency and accuracy is particularly beneficial for downstream tasks where motif statistics are crucial (e.g., Milo et al. [2002]).

## J  Details of the power iteration method

To estimate the leading $r$ ($r \geq 1$) eigenpairs of a matrix $A$, we use the power iteration method iteratively, combined with matrix deflation. Beginning with $A_0 = A$, each step $k$ (from 1 to $r$) estimates the dominant eigenpair $(\hat{\lambda}_k, \hat{v}_k)$ of $A_{k-1}$ using the power iteration method. After normalization of $\hat{v}_k$, the matrix is deflated as $A_k = A_{k-1} - \hat{\lambda}_k \hat{v}_k \hat{v}_k^\top$. Once all $r$ eigenpairs are computed, an approximation $\tilde{P}$ is reconstructed as $\tilde{P} = \sum_{k=1}^{r} \hat{\lambda}_k \hat{v}_k \hat{v}_k^\top$. Finally, the probability connection matrix $\hat{P}$ is

obtained via element-wise thresholding:

$$p_{ij} = \min\left(1, \max(0, \tilde{p}_{ij})\right),$$

ensuring all entries lie within the probability range $[0, 1]$. The complete algorithm is summarized in Algorithm 5.

---

**Algorithm 5** Iterative Power Iteration.

---

**Require:** Adjacency matrix $A$, rank $r$, maximum iterations $N$, tolerance $\epsilon$.
**Ensure:** Estimated probability connection matrix $\hat{P}$.
1: Initialize $A_0 \leftarrow A$.
2: **for** $k = 1$ to $r$ **do**
3:    Initialize vector $\mathbf{x}_0$ of length $n$ with all ones and normalize: $\mathbf{x}_0 \leftarrow \mathbf{x}_0/\|\mathbf{x}_0\|_2$.
4:    **for** $i = 1$ to $N$ **do**
5:        Update $\mathbf{x}_i \leftarrow A_{k-1}\mathbf{x}_{i-1}$ and normalize: $\mathbf{x}_i \leftarrow \mathbf{x}_i/\|\mathbf{x}_i\|_2$.
6:        **if** $\|\mathbf{x}_i - \mathbf{x}_{i-1}\|_2 < \epsilon$ **then**
7:            **break**
8:        **end if**
9:    **end for**
10:    Compute eigenvalue $\hat{\lambda}_k \leftarrow \mathbf{x}_i^\top A_{k-1}\mathbf{x}_i$ and eigenvector $\hat{v}_k \leftarrow \mathbf{x}_i$.
11:    Deflate the matrix: $A_k \leftarrow A_{k-1} - \hat{\lambda}_k \hat{v}_k \hat{v}_k^\top$.
12: **end for**
13: Compute $\tilde{P} = \sum_{k=1}^{r} \hat{\lambda}_k \hat{v}_k \hat{v}_k^\top$ and threshold element-wise: $\hat{p}_{ij} = \min\left(1, \max(0, \tilde{p}_{ij})\right)$.
14: **Return** $\hat{P}$.

---

For the experiments, we set the maximum number of iterations to 500 and the convergence threshold to $10^{-6}$. Tables 14 and 15 summarize the actual number of iterations for all scenarios discussed in this paper. In some cases, the power iteration does not converge within 500 iterations.

Table 14: Number of iterations for dense settings across 100 independent trials.

| ID | $\hat{\lambda}_1$ Iterations | Std. Dev. | $\hat{\lambda}_2$ Iterations | Std. Dev. | $\hat{\lambda}_3$ Iterations | Std. Dev. |
|----|------|--------|-------|--------|------|--------|
| 1 | 6 | 0 | | | | |
| 2 | 5 | 0 | | | | |
| 3 | 5.01 | 0.0995 | | | | |
| 4 | 9 | 0 | 17.99 | 0.5744 | | |
| 5 | 6 | 0 | 500 | 0 | | |
| 6 | 15.34 | 0.6200 | 8 | 0 | | |
| 7 | 27.83 | 1.8871 | 15.1 | 0.7416 | 8 | 0 |

Table 15: Number of iterations for sparse settings across 100 independent trials.

| ID | $\hat{\lambda}_1$ Iterations | Std. Dev. | $\hat{\lambda}_2$ Iterations | Std. Dev. |
|----|------|--------|-------|--------|
| 2 | 13 | 0 | | |
| 3 | 17.15 | 0.3571 | | |
| 4 | 31.91 | 1.8713 | 500 | 0 |
| 5 | 17.98 | 0.3995 | 500 | 0 |

# K Proofs

*Proof of Theorem 3.1.* By (2) and (3), we have

$$\sup_i \left| G_1(U_i) - \frac{1}{c(n-1)} d_i \right| = O_p\left(\sqrt{\frac{\log(n)}{n}}\right), \tag{12}$$

where $c = \lambda_1 \int_0^1 G_1(u)\, du$.

Using the property of U-statistics (see, for example, Theorem 4.2.1 in Korolyuk [2013]), we have

$$\frac{1}{n(n-1)} \sum_{i,j:i\neq j} f(U_i, U_j) = \mathbb{E}f(U_i, U_j) + O_p(n^{-1/2}). \tag{13}$$

Moreover, note that

$$\mathbb{E}\left(\left(\frac{1}{n(n-1)} \sum_{i,j:i\neq j} E_{ij} - \frac{1}{n(n-1)} \sum_{i,j:i\neq j} f(U_i, U_j)\right)^2 \bigg| U_1, \ldots, U_n\right) \tag{14}$$

$$\lesssim \frac{1}{n^4} \sum_{i_1,i_2,j_1,j_2} \mathbb{E}\left((E_{i_1 j_1} - f(U_{i_1}, U_{j_1}))(E_{i_2 j_2} - f(U_{i_2}, U_{j_2}))\bigg| U_1, \ldots, U_n\right) \tag{15}$$

$$\lesssim \frac{1}{n^4} \sum_{i_1,i_2} \mathbb{E}\left((E_{i_1 j_1} - f(U_{i_1}, U_{j_1}))^2 \bigg| U_1, \ldots, U_n\right) = O\left(\frac{1}{n^2}\right), \tag{16}$$

where the second inequality follows from the fact that the terms are nonzero only when $i_1 = i_2, j_1 = j_2$, and the last equality is due to the boundedness of each term.

Combining (13) and (14), we obtain

$$\frac{1}{n(n-1)} \sum_{i,j:i\neq j} E_{ij} = \lambda_1 \left(\int_0^1 G_1(u)\, du\right)^2 + O_p(n^{-1/2}). \tag{17}$$

Similarly,

$$\frac{1}{n(n-1)^3} \sum_{i,j:i\neq j} d_i d_j = \lambda_1^2 \left(\int_0^1 G_1(u)\, du\right)^4 + O_p(n^{-1/2}). \tag{18}$$

Combining (17) and (18), we have

$$(n-1)^2 \frac{\sum_{i,j:i\neq j} E_{ij}}{\sum_{i,j:i\neq j} d_i d_j} = \frac{1}{\lambda_1 \left(\int_0^1 G_1(u)\, du\right)^2} + O_p(n^{-1/2}).$$

Thus,

$$\sup_i \left| G_1(U_i) - \sqrt{\frac{\sum_{i,j:i\neq j} E_{ij}}{\sum_{i,j:i\neq j} d_i d_j}} \frac{d_i}{\sqrt{\lambda_1}} \right| \tag{19}$$

$$\leq \sup_i \left| G_1(U_i) - \frac{1}{c(n-1)} d_i \right| + \sup_i \left| \sqrt{\frac{\sum_{i,j:i\neq j} E_{ij}}{\sum_{i,j:i\neq j} d_i d_j}} \frac{d_i}{\sqrt{\lambda_1}} - \frac{1}{c(n-1)} d_i \right| \tag{20}$$

$$\leq \sup_i \left| G_1(U_i) - \frac{1}{c(n-1)} d_i \right| + \left| \sqrt{(n-1)^2 \frac{\sum_{i,j:i\neq j} E_{ij}}{\sum_{i,j:i\neq j} d_i d_j}} \frac{1}{\sqrt{\lambda_1}} - \frac{1}{\lambda_1 \int_0^1 G_1(u) du} \right| \tag{21}$$

$$= O_p\left(\sqrt{\frac{\log(n)}{n}}\right). \tag{22}$$

By the definition of the graphon function, $\sup_{u_1, u_2 \in [0,1]} \lambda_1 G_1(u_1) G_1(u_2) \leq 1$, and thus $\sup_{u \in [0,1]} \sqrt{\lambda_1} G_1(u) \leq 1$.

For $c_1 = \frac{\sum_{i,j:i\neq j} E_{ij}}{\sum_{i,j:i\neq j} d_i d_j}$, we have

$$
\begin{aligned}
\sup_{i,j} |\hat{p}_{ij} - p_{ij}| &\leq \sup_{i,j} |c_1 d_i d_j - \lambda_1 G_1(U_i) G_1(U_j)| \\
&\leq \sup_{i,j} \left| \sqrt{c_1} d_i - \sqrt{\lambda_1} G_1(U_i) \right| \sqrt{c_1} d_j \\
&\quad + \sup_{i,j} \left| \sqrt{c_1} d_j - \sqrt{\lambda_1} G_1(U_j) \right| \sqrt{\lambda_1} G_1(U_i) \\
&= O_p \left( \sqrt{\frac{\log(n)}{n}} \right).
\end{aligned}
$$

$\square$

*Proof of Theorem 3.2.* It suffices to show that

$$
\sup_{u \in [0,1]} \left| G_1^\dagger(u) - \frac{1}{(n-1)\lambda_1 \int_0^1 G_1(v) dv} h(u) \right| \overset{a.s.}{\to} 0, \tag{23}
$$

$$
\sup_{u \in [0,1]} \left| G_1^\dagger(u) - \frac{1}{(n-1)\lambda_1 \int_0^1 G_1(v) dv} h(u) \right| = O_p \left( \sqrt{\frac{\log(n)}{n}} \right), \tag{24}
$$

and the subsequent steps can then be obtained by following the similar proof of Theorem 3.1, transitioning from (12) to (19) by replacing $(n-1)\lambda_1 \int_0^1 G_1(v) dv$ with $\sqrt{\frac{\sum_{i,j:i\neq j} E_{ij}}{\lambda_1 \sum_{i,j:i\neq j} d_i d_j}}$, and modifying the argument from taking the maximum over all $U_i$ to taking the supremum over all $u \in [0,1]$. To show (23) and (24), we consider the following two steps.

**(Step 1.)** In this step, we prove that

$$
\sup_{u \in \{1,2,\cdots,n\}} \left| \frac{h\left(\frac{u}{n+1}\right)}{(n-1)\lambda_1 \int_0^1 G_1(v) dv} - G_1^\dagger\left(\frac{u}{n+1}\right) \right| \overset{a.s.}{\to} 0,
$$

and

$$
\sup_{u \in \{1,2,\cdots,n\}} \left| \frac{h\left(\frac{u}{n+1}\right)}{(n-1)\lambda_1 \int_0^1 G_1(v) dv} - G_1^\dagger\left(\frac{u}{n+1}\right) \right| = O_p \left( \sqrt{\frac{\log(n)}{n}} \right).
$$

Let $U_{(1)}, \cdots, U_{(n)}$ denote the rearrangement of $U_1, \cdots, U_n \overset{i.i.d.}{\sim}$ Uniform$(0,1)$ such that $U_{(1)} \leq \cdots \leq U_{(n)}$. By Lemma L.2, we have

$$
\sup_{i=1,\cdots,n} |U_{(i)} - i/(n+1)| \overset{a.s.}{\to} 0.
$$

By Kawohl [2006] (Chapter II.2), the rearrangement function $G_1^\dagger$ is Lipschitz continuous with constant $M$ as long as $G_1$ is Lipschitz continuous with constant $M$. As a consequence,

$$
\sup_{i=1,\cdots,n} |G_1^\dagger(U_{(i)}) - G_1^\dagger(i/(n+1))| \leq M \sup_{i=1,\cdots,n} |U_{(i)} - i/(n+1)| \overset{a.s.}{\to} 0. \tag{25}
$$

Moreover, using the proof of Lemma 1 in Chan and Airoldi [2014], we have

$$
\sup_{i=1,\cdots,n} |U_{(i)} - i/(n+1)| = O_p \left( \sqrt{\frac{\log(n)}{n}} \right),
$$

which also shows that

$$
\sup_{i=1,\cdots,n} |G_1^\dagger(U_{(i)}) - G_1^\dagger(i/(n+1))| = O_p \left( \sqrt{\frac{\log(n)}{n}} \right). \tag{26}
$$

By definition, for $i = 1, \cdots, n$, $h(i/(n+1)) = d_{\sigma(i)}$. By (12) (more precisely, the similar argument of (12) applied to $G^\dagger$), (25), and Lemma L.3, we have

$$\sup_{i \in \{1,2,\cdots,n\}} \left| \frac{h\left(\frac{i}{n+1}\right)}{(n-1)\lambda_1 \int_0^1 G_1(v)dv} - G_1^\dagger\left(\frac{i}{n+1}\right) \right| \overset{a.s.}{\to} 0.$$

Similarly, via (12), (26), and Lemma L.3, we have

$$\sup_{i \in \{1,2,\cdots,n\}} \left| \frac{h\left(\frac{i}{n+1}\right)}{(n-1)\lambda_1 \int_0^1 G_1(v)dv} - G_1^\dagger\left(\frac{i}{n+1}\right) \right| = O_p\left(\sqrt{\frac{\log(n)}{n}}\right).$$

**(Step 2.)** In this step, we prove (23). We note that

$$\sup_{u \in [0,1/(n+1)]} \left| G_1^\dagger(u) - \frac{1}{(n-1)\lambda_1 \int_0^1 G_1(v)dv} h(u) \right|$$

$$\leq \left| G_1^\dagger\left(\frac{1}{n+1}\right) - \frac{h(1/(n+1))}{(n-1)\lambda_1 \int_0^1 G_1(v)dv} \right|$$

$$+ \sup_{u \in [0,1/(n+1)]} \left| G_1^\dagger\left(\frac{1}{n+1}\right) - G_1^\dagger(u) \right|$$

$$\leq \left| G_1^\dagger\left(\frac{1}{n+1}\right) - \frac{h(1/(n+1))}{(n-1)\lambda_1 \int_0^1 G_1(v)dv} \right| + \frac{M}{n+1}$$

$$\overset{a.s.}{\to} 0, \quad \text{and} \quad O_p\left(\sqrt{\frac{\log(n)}{n}}\right).$$

Similarly, we have

$$\sup_{u \in [n/(n+1),1]} \left| G_1^\dagger(u) - \frac{1}{(n-1)\lambda_1 \int_0^1 G_1(v)dv} h(u) \right| \overset{a.s.}{\to} 0, \quad \text{and} \quad O_p\left(\sqrt{\frac{\log(n)}{n}}\right).$$

For $u \in (1/(n+1), n/(n+1))$, let $k = \lfloor u(n+1) \rfloor$, then

$$\left| G_1^\dagger(u) - \frac{h(u)}{(n-1)\lambda_1 \int_0^1 G_1(v)dv} \right|$$

$$\leq (k+1-u(n+1)) \left| G_1^\dagger(u) - G_1^\dagger\left(\frac{k}{n+1}\right) \right|$$

$$+ (k+1-u(n+1)) \left| G_1^\dagger\left(\frac{k}{n+1}\right) - \frac{h\left(\frac{k}{n+1}\right)}{(n-1)\lambda_1 \int_0^1 G_1(v)dv} \right|$$

$$+ (u(n+1)-k) \left| G_1^\dagger\left(\frac{k+1}{n+1}\right) - \frac{h\left(\frac{k+1}{n+1}\right)}{(n-1)\lambda_1 \int_0^1 G_1(v)dv} \right|$$

$$+ (u(n+1)-k) \left| G_1^\dagger(u) - G_1^\dagger\left(\frac{k+1}{n+1}\right) \right|$$

$$\leq \frac{M}{n+1} + \sup_{i \in \{1,2,\cdots,n\}} \left| \frac{h\left(\frac{i}{n+1}\right)}{(n-1)\lambda_1 \int_0^1 G_1(v)dv} - G_1^\dagger\left(\frac{i}{n+1}\right) \right|.$$

Therefore, by the result from **(Step 1)**,

$$\sup_{u \in [1/(n+1),n/(n+1)]} \left| G_1^\dagger(u) - \frac{1}{(n-1)\lambda_1 \int_0^1 G_1(v)dv} h(u) \right| = O_p\left(\sqrt{\frac{\log(n)}{n}}\right).$$

Finally, combining the results of the steps yields the proof of Theorem 3.2. □

*Proof of Theorem 3.6.* Without loss of generality, we assume that $\int_0^1 G_k(u)\,du \geq 0$ for $1 \leq k \leq r$. If $\int_0^1 G_k(u)\,du \leq 0$, we can replace $G_k$ with $-G_k$.

For $i = 1, \ldots, n$, recall that

$$L_i^{(1)} = \sum_{i_1} A_{ii_1},$$

$$L_i^{(a)} = \sum_{\substack{i_1, \ldots, i_a \text{ distinct,} \\ i_k \neq i, 1 \leq k \leq a}} E_{ii_1} \prod_{j=2}^{a} E_{i_{j-1}i_j} \quad \text{for } a \geq 2,$$

$$C_i^{(a)} = \sum_{\substack{i_1, \ldots, i_{a-1} \text{ distinct,} \\ i_k \neq i, 1 \leq k \leq a-1}} E_{ii_1} E_{i_{a-1}i} \prod_{j=2}^{a-1} E_{i_{j-1}i_j} \quad \text{for } a \geq 3.$$

Note that $\mathbb{P}(E_{ij} = 1 \mid U_i, U_j) = \sum_{k=1}^r \lambda_k G_k(U_i) G_k(U_j)$ and $\int_0^1 G_i^2(u)\,du = 1$ for $1 \leq i \leq r$. We then have

$$\frac{1}{\prod_{j=1}^{a}(n-j)} \mathbb{E}(L_i^{(a)} \mid U_i) = \sum_{k=1}^{r} \lambda_k^a G_k(U_i) \int_0^1 G_k(u)\,du \quad \text{for } 1 \leq a \leq r,$$

$$\frac{1}{\prod_{j=1}^{a-1}(n-j)} \mathbb{E}(C_i^{(a)} \mid U_i) = \sum_{k=1}^{r} \lambda_k^a G_k^2(U_i) \quad \text{for } 3 \leq a \leq r+2. \tag{27}$$

We prove the theorem in two steps.

**(Step 1.)** We first show that

$$\max_{1 \leq k \leq r} |\hat{\lambda}_k - \lambda_k| = O_p(n^{-1/2}), \quad \max_{1 \leq k \leq r} \left| y_k - \int_0^1 G_k(u)\,du \right| = O_p(n^{-1/2}).$$

By (27), we have

$$\frac{1}{\prod_{j=1}^{a}(n-j)} \mathbb{E}(L_i^{(a)}) = \sum_{k=1}^{r} \lambda_k^a \left( \int_0^1 G_k(u)\,du \right)^2 \quad \text{for } 1 \leq a \leq r,$$

$$\frac{1}{\prod_{j=1}^{a-1}(n-j)} \mathbb{E}(C_i^{(a)}) = \sum_{k=1}^{r} \lambda_k^a \quad \text{for } 3 \leq a \leq r+2. \tag{28}$$

Moreover, by the implicit function theorem, the system of equations (28) in terms of $\lambda_k$ and $\left( \int_0^1 G_k(u)\,du \right)^2$ for $1 \leq k \leq r$ has a unique solution if

$$\begin{vmatrix} \lambda_1^2 & \lambda_2^2 & \cdots & \lambda_r^2 \\ \vdots & \vdots & \ddots & \vdots \\ \lambda_1^{r+1} & \lambda_2^{r+1} & \cdots & \lambda_r^{r+1} \end{vmatrix} \neq 0, \quad \begin{vmatrix} \lambda_1 & \lambda_2 & \cdots & \lambda_r \\ \vdots & \vdots & \ddots & \vdots \\ \lambda_1^r & \lambda_2^r & \cdots & \lambda_r^r \end{vmatrix} \neq 0. \tag{29}$$

This condition is satisfied under Assumption 3.5, which assumes $\lambda_k > 0$ for $1 \leq k \leq r$ and $\lambda_i \neq \lambda_j$ for $i \neq j$.

By Lemma L.4, we have

$$\frac{1}{\prod_{j=0}^{a}(n-j)} \sum_{i=1}^{n} \left( L_i^{(a)} - \mathbb{E}(L_i^{(a)}) \right) = O_p(n^{-1/2}) \quad \text{for } 1 \leq a \leq r,$$

$$\frac{1}{\prod_{j=0}^{a-1}(n-j)} \sum_{i=1}^{n} \left( C_i^{(a)} - \mathbb{E}(C_i^{(a)}) \right) = O_p(n^{-1/2}) \quad \text{for } 3 \leq a \leq r+2.$$

By Lemma L.5, we have

$$\max_{1 \leq k \leq r} |\hat{\lambda}_k - \lambda_k| = O_p(n^{-1/2}).$$

By Lemma L.6, we have

$$\max_{1 \le k \le r} \left| y_k - \int_0^1 G_k(u)\,du \right| = O_p(n^{-1/2}).$$

Note that there is no ambiguity in the square root since we assume $\int_0^1 G_i(u)\,du \ge 0$ for $i = 1, 2$.

**(Step 2.)** In this step, we prove that

$$\sup_{i,j} |\hat{p}_{ij} - p_{ij}| = O_p\left(\sqrt{\frac{\log n}{n}}\right).$$

Recall that $(G_1(U_i), \ldots, G_r(U_i))$ is estimated by solving the system of equations with respect to $(\hat{G}_1(U_i), \ldots, \hat{G}_r(U_i))$:

$$\frac{1}{\prod_{j=1}^a (n-j)} L_i^{(a)} = \sum_{k=1}^r \hat{\lambda}_k^a y_k \hat{G}_k(U_i) \quad \text{for } 1 \le a \le r,$$

where $\hat{\lambda}_k^a$ and $y_k$ are defined in (8). For this linear system, we have

$$\max_i \max_k |\hat{G}_k(U_i) - G_k(U_i)| = O_p\left(\sqrt{\frac{\log n}{n}}\right), \tag{30}$$

provided that

$$\max_i \max_a \frac{|L_i^{(a)} - \mathbb{E}(L_i^{(a)} \mid U_i)|}{\prod_{j=1}^a (n-j)} = O_p\left(\sqrt{\frac{\log n}{n}}\right),$$

which is guaranteed by Lemma L.7.

By (10), for every $1 \le k \le r$, we have

$$\tilde{G}_k(U_i) - \hat{G}_k(U_i) = \frac{\hat{G}_k(U_i)}{\sqrt{\frac{1}{n} \sum_{i=1}^n \hat{G}_k^2(U_i)}} - \hat{G}_k(U_i) = \hat{G}_k(U_i)\left(\frac{1}{\sqrt{\frac{1}{n} \sum_{i=1}^n \hat{G}_k^2(U_i)}} - 1\right). \tag{31}$$

Since $U_i$'s are i.i.d., we have

$$\frac{1}{n} \sum_{i=1}^n G_k^2(U_i) - 1 = O_p(n^{-1/2}).$$

Thus,

$$\frac{1}{n} \sum_{i=1}^n \hat{G}_k^2(U_i) - 1 = \frac{1}{n} \sum_{i=1}^n \hat{G}_k^2(U_i) - \frac{1}{n} \sum_{i=1}^n G_k^2(U_i) + \frac{1}{n} \sum_{i=1}^n G_k^2(U_i) - 1 = O_p\left(\sqrt{\frac{\log n}{n}}\right),$$

which implies

$$\frac{1}{\sqrt{\frac{1}{n} \sum_{i=1}^n \hat{G}_k^2(U_i)}} - 1 = O_p\left(\sqrt{\frac{\log n}{n}}\right). \tag{32}$$

By Assumption 3.5, $G_k$ is bounded by $K$. Combining (32), (31), (30), and noting that $r = O(1)$, we have

$$\max_k \max_i |\tilde{G}_k(U_i) - \hat{G}_k(U_i)| = O_p\left(\sqrt{\frac{\log n}{n}}\right).$$

Therefore,

$$\max_k \max_i |\tilde{G}_k(U_i) - G_k(U_i)| = O_p\left(\sqrt{\frac{\log n}{n}}\right).$$

As a result, for the estimation of connection probabilities, we have

$$\sup_{i,j} |\hat{p}_{ij} - p_{ij}| = \sup_{i,j} \left| \left[ 1 \wedge \left( 0 \vee \left( \sum_{k=1}^{r} \hat{\lambda}_k \tilde{G}_k(U_i) \tilde{G}_k(U_j) \right) \right) \right] - \left( \sum_{k=1}^{r} \lambda_k G_k(U_i) G_k(U_j) \right) \right|$$

$$= O_p \left( \sqrt{\frac{\log n}{n}} \right).$$

$\square$

*Proof of Lemma D.2.* We show that

$$\max_{1 \leq i \leq n} \max_{1 \leq a \leq r} \left| \frac{1}{\prod_{j=1}^{a}(n-j)} \left( \tilde{L}_i^{(a)} - L_i^{(a)} \right) \right| = o_p \left( \frac{1}{\sqrt{n}} \right),$$

as the result for $\tilde{C}_i^{(a)}$ follows similarly.

By definition, we have

$$\tilde{L}_i^{(a)} - L_i^{(a)} = \sum_{i_1,\ldots,i_a \in \mathcal{M}} E_{i,i_1} \prod_{j=2}^{a} E_{i_{j-1},i_j},$$

where

$$\mathcal{M} = \{\text{At least two of the values } i, i_1, \ldots, i_a \text{ are identical}\}.$$

Therefore,

$$\frac{1}{\prod_{j=1}^{a}(n-j)} \left| \tilde{L}_i^{(a)} - L_i^{(a)} \right| \leq \frac{1}{\prod_{j=1}^{a}(n-j)} \sum_{i_1,\ldots,i_a \in \mathcal{M}} 1 = \frac{O(n^{a-1})}{\prod_{j=1}^{a}(n-j)}.$$

As a result, we have

$$\max_{1 \leq i \leq n} \max_{1 \leq a \leq r} \frac{1}{\prod_{j=1}^{a}(n-j)} \left| \tilde{L}_i^{(a)} - L_i^{(a)} \right| \leq \frac{O(n^{a-1})}{\prod_{j=1}^{a}(n-j)} = O_p \left( \frac{1}{n} \right).$$

$\square$

# L   Technical Lemmas

**Lemma L.1.** *For the rank-2 model with $f(u,v) = \lambda_1 G_1(u) G_1(v) + \lambda_2 G_2(u) G_2(v)$, where $G_1, G_2$ are bounded by a constant $M > 0$, we have*

$$\sup_{i=1,\ldots,n} \frac{|d_i - \mathbb{E}(d_i \mid U_i)|}{n-1} = O_p \left( \sqrt{\frac{\log(n)}{n}} \right),$$

*where $d_i$ is the degree of the $i$-th node. Note that the model reduces to a rank-1 model when $\lambda_2 = 0$.*

*Proof.* We first note that

$$\sup_{i=1,\ldots,n} \left| \frac{1}{n-1} \sum_{j:j \neq i} (\lambda_1 G_1(U_i) G_1(U_j) + \lambda_2 G_2(U_i) G_2(U_j)) \right.$$

$$\left. -\lambda_1 G_1(U_i) \int_0^1 G_1(u) \, du - \lambda_2 G_2(U_i) \int_0^1 G_2(u) \, du \right|$$

$$\leq \lambda_1 M \left( \left| \frac{1}{n-1} \sum_{j=1}^{n} G_1(U_j) - \int_0^1 G_1(u) \, du \right| + \frac{1}{n-1} M \right)$$

$$+ \lambda_2 M \left( \left| \frac{1}{n-1} \sum_{j=1}^{n} G_2(U_j) - \int_0^1 G_2(u) \, du \right| + \frac{1}{n-1} M \right) = O_p(n^{-1/2}),$$

where the last result follows from Slutsky's theorem.

It now suffices to show that

$$\sup_{i=1,\ldots,n} \left| \frac{1}{n-1} \sum_{j:j\neq i} \left( I\left( U_{ij} \leq \lambda_1 G_1(U_i)G_1(U_j) + \lambda_2 G_2(U_i)G_2(U_j) \right) \right. \right. \tag{33}$$

$$\left. \left. -\lambda_1 G_1(U_i)G_1(U_j) - \lambda_2 G_2(U_i)G_2(U_j) \right) \right| = O_p\left( \sqrt{\frac{\log(n)}{n}} \right), \tag{34}$$

where $U_{ij}, i \leq j$ are i.i.d. uniformly distributed random variables on $[0,1]$, and $U_{ji} = U_{ij}$ for $i > j$.

Let

$$Z_i = \frac{1}{n-1} \sum_{j=1}^{n} \left( I\left( U_{ij} \leq \lambda_1 G_1(U_i)G_1(U_j) + \lambda_2 G_2(U_i)G_2(U_j) \right) -\lambda_1 G_1(U_i)G_1(U_j) - \lambda_2 G_2(U_i)G_2(U_j) \right).$$

By Hoeffding's inequality (Theorem 2.6.2 of Vershynin [2018]), for any $t > 0$, we have

$$\mathbb{P}\left( \sqrt{n}|Z_i| > t \mid U_1, \ldots, U_n \right) \leq 2\exp(-ct^2),$$

where $c > 0$ is an absolute constant.

Taking expectations, we get

$$\mathbb{P}\left( \sqrt{n}|Z_i| > t \right) = \mathbb{E}\left( \mathbb{P}\left( \sqrt{n}|Z_i| > t \mid U_1, \ldots, U_n \right) \right) \leq 2\exp(-ct^2).$$

As a result, $\sqrt{n}Z_i$ are sub-Gaussian random variables. Then, using standard bounds for maxima of sub-Gaussian variables, we have

$$\mathbb{E}\max_{i=1,\ldots,n} |Z_i| = O\left( \sqrt{\frac{\log(n)}{n}} \right),$$

which implies that

$$\max_{i=1,\ldots,n} |Z_i| = O_p\left( \sqrt{\frac{\log(n)}{n}} \right).$$

$\square$

**Lemma L.2.** *Suppose that* $U_i \overset{i.i.d.}{\sim} Uniform(0,1)$, *for* $i = 1, \ldots, n$. *Let* $U_{(i)}$ *denote the* $i$-*th smallest value among* $U_1, \ldots, U_n$, *i.e.,* $U_{(1)} \leq U_{(2)} \leq \cdots \leq U_{(n)}$. *Then*

$$\sup_i \left| U_{(i)} - \frac{i}{n+1} \right| \overset{a.s.}{\to} 0.$$

*Proof.* It is well-known that $U_{(i)} \sim \text{Beta}(i, n-i+1)$, with probability density function

$$p(x) = \frac{x^{i-1}(1-x)^{n-i}}{\int_0^1 x^{i-1}(1-x)^{n-i}\,dx}.$$

We now proceed to bound the tail probability for any $\varepsilon > 0$ using Markov's inequality. First, we have

$$\mathbb{P}\left( \left| U_{(i)} - \frac{i}{n+1} \right| \geq \varepsilon \right) \leq \frac{1}{\varepsilon^6} \mathbb{E}\left| U_{(i)} - \frac{i}{n+1} \right|^6$$

$$= \frac{1}{\varepsilon^6} \frac{5i(n-i+1)A}{(n+1)^6(n+2)(n+3)(n+4)(n+5)(n+6)}$$

$$\leq \frac{1}{\varepsilon^6} \frac{5n^2 A}{(n+1)^{11}},$$

where

$$A = 24(n-i+1)^4 + 2i(n-i+1)^3(13n-13i+1) + i^2(n-i+1)^2(24 - 8(n-i+1) + 3(n-i+1)^2)$$

$$+2i^3(n-i+1)^2(3(n-i+1)^2 - 4(n-i+1) - 12) + i^4(24 + 26(n-i+1) + 3(n-i+1)^2)).$$

We note that $A \le 12n^6 + 36n^5 + 24n^4 \le 72n^6$. Hence, we have

$$\sum_{n=1}^{\infty} \mathbb{P}\left(\sup_i \left|U_{(i)} - \frac{i}{n+1}\right| \ge \varepsilon\right) \le \sum_{n=1}^{\infty}\sum_{i=1}^{n} \mathbb{P}\left(\left|U_{(i)} - \frac{i}{n+1}\right| \ge \varepsilon\right)$$

$$\le \frac{1}{\varepsilon^6}\sum_{n=1}^{\infty}\sum_{i=1}^{n}\frac{360n^8}{(n+1)^{11}}$$

$$\le \frac{360}{\varepsilon^6}\sum_{n=1}^{\infty}\frac{1}{n^2} < \infty.$$

Therefore, by the Borel-Cantelli lemma, the result follows. $\qquad\square$

**Lemma L.3.** *Let $G(u)$, for $u \in [0,1]$, be a monotonically non-decreasing, Lipschitz continuous function with Lipschitz constant $L > 0$. Let $a_i := G(i/(n+1)), i = 1, \ldots, n$. Suppose that there exists a sequence of random variables $b_1, \ldots, b_n$ such that $\sup_{i=1,\ldots,n} |b_i - a_i| \overset{a.s.}{\to} 0$. Let $\alpha$ be a one-to-one permutation such that $b_{\alpha(1)} \le b_{\alpha(2)} \le \cdots \le b_{\alpha(n)}$. Let $\hat{a}_i := b_{\alpha(i)}$. Then we have*

$$\sup_i |\hat{a}_i - a_i| \overset{a.s.}{\to} 0.$$

*Moreover, if $\sup_{i=1,\ldots,n} |b_i - a_i| = O_p(g_n)$, then $\sup_i |\hat{a}_i - a_i| = O_p(g_n)$ for some $g_n = o(1)$, with $ng_n \to \infty$.*

*Proof.* Let $M_n = \sup_{i=1,\ldots,n} |b_i - a_i|$, then $M_n \overset{a.s.}{\to} 0$. Assume without loss of generality that $1/n = o_{a.s.}(M_n)$. Let $K_n$ be a non-negative random variable such that $K_n \overset{a.s.}{\to} 0$, $3M_n \le K_n \le 4M_n$, and $1/n = o_{a.s.}(K_n)$. For any $i = 1, \ldots, n$, we have

$$|\hat{a}_i - a_i| = |b_{\alpha(i)} - a_i| \le |a_{\alpha(i)} - a_i| + M_n.$$

First, consider the case where $\alpha(i) \ge i$. Assume, for the sake of contradiction, that $a_{\alpha(i)} - a_i > K_n$. Then for $j = 1, 2, \ldots, i+1$, we derive that

$$b_{\alpha(i)} \ge a_{\alpha(i)} - M_n > a_j - \frac{L}{n+1} + K_n - M_n \ge b_j - \frac{L}{n+1} + K_n - 2M_n,$$

where for the second inequality, we use the monotonicity of $G(u)$. By the construction of $K_n$, with probability 1, when $n$ is sufficiently large, we have $b_{\alpha(i)} > b_j$ for $j = 1, 2, \ldots, i+1$. This implies that there are at least $i+1$ values smaller than $b_{\alpha(i)}$, which contradicts the definition of $\alpha$. Therefore, $a_{\alpha(i)} - a_i \le K_n$.

Similarly, for the case where $\alpha(i) \le i$, we have $a_{\alpha(i)} - a_i \ge -K_n$.

We thus conclude that

$$\sup_i |\hat{a}_i - a_i| = O_{a.s.}(K_n) + M_n \overset{a.s.}{\to} 0.$$

The statement for $O_p$ follows from the same argument. $\qquad\square$

**Lemma L.4.** *Under the assumptions of Theorem 3.6, we have*

$$\frac{1}{\prod_{j=0}^{a}(n-j)}\sum_{i=1}^{n}\left(L_i^{(a)} - \mathbb{E}(L_i^{(a)})\right) = O_p(n^{-1/2}) \quad \text{for } 1 \le a \le r,$$

$$\frac{1}{\prod_{j=0}^{a-1}(n-j)}\sum_{i=1}^{n}\left(C_i^{(a)} - \mathbb{E}(C_i^{(a)})\right) = O_p(n^{-1/2}) \quad \text{for } 3 \le a \le r+2,$$

*where $L_i^{(a)}, C_i^{(a)}$ are defined in Section 3.2.*

*Proof.* We only show that

$$\frac{1}{\prod_{j=0}^{a}(n-j)}\sum_{i=1}^{n}\left(L_i^{(a)} - \mathbb{E}(L_i^{(a)})\right) = O_p(n^{-1/2}) \quad \text{for } 1 \le a \le r,$$

as the results for $C_i^{(a)}$ follow similarly.

Note that $\mathbb{E}(E_{ij}|U_i, U_j) = f(U_i, U_j)$, and that $E_{ij}$ is conditionally independent of $E_{i_1, j_1}$ when $(i, j) \neq (i_1, j_1)$. Then we derive that

$$\frac{1}{\left(\prod_{j=1}^{a}(n-j)\right)^2}\mathbb{E}\left[\left(\sum_{i=1}^{n} L_i^{(a)} - \sum_{i=1}^{n}\mathbb{E}(L_i^{(a)} \mid U_1, \ldots, U_n)\right)^2 \Big| U_1, \ldots, U_n\right]$$

$$\lesssim \frac{1}{n^{2a+2}} \sum_{i,i_1,\ldots,i_a,k,k_1,\ldots,k_a} \mathbb{E}\left[\left(E_{ii_1}\prod_{j=2}^{a} E_{i_{j-1}i_j} - f(U_i, U_{i_1})\prod_{j=2}^{a} f(U_{i_{j-1}}, U_{i_j})\right)\right.$$

$$\left.\left(E_{kk_1}\prod_{j=2}^{a} E_{k_{j-1}k_j} - f(U_k, U_{k_1})\prod_{j=2}^{a} f(U_{k_{j-1}}, U_{k_j})\right)\Big| U_1, \ldots, U_n\right]$$

$$\lesssim \frac{n^{2a}}{n^{2a+2}} = \frac{1}{n^2}.$$

Since $\sum_{i=1}^{n} L_i^{(a)} \leq \prod_{j=0}^{a}(n-j)$, we have

$$\frac{1}{\prod_{j=0}^{a}(n-j)}\sum_{i=1}^{n}\left(L_i^{(a)} - \mathbb{E}(L_i^{(a)}|U_1,\ldots,U_n)\right) = O_p\left(\frac{1}{n}\right). \tag{35}$$

Moreover, by the property of U-statistics (see, for example, Theorem 4.2.1 in Korolyuk [2013]), we have

$$\frac{\sum_{i,i_1,\ldots,i_a} f(U_i, U_{i_1})\prod_{j=2}^{a} f(U_{i_{j-1}}, U_{i_j})}{\prod_{j=0}^{a}(n-j)} = \frac{\mathbb{E}\sum_{i,i_1,\ldots,i_a} f(U_i, U_{i_1})\prod_{j=2}^{a} f(U_{i_{j-1}}, U_{i_j})}{\prod_{j=0}^{a}(n-j)} + O_p(n^{-1/2}). \tag{36}$$

Note that

$$\sum_{i=1}^{n}\mathbb{E}(L_i^{(a)}|U_1,\ldots,U_n) = \sum_{i,i_1,\ldots,i_a} f(U_i, U_{i_1})\prod_{j=2}^{a} f(U_{i_{j-1}}, U_{i_j}).$$

Then the result follows by combining (36) with (35). $\qquad\square$

**Lemma L.5.** *Suppose that $x_1, \ldots, x_r$ are $r$ real numbers satisfying $|x_1| > |x_2| > \cdots > |x_r| > 0$. Let $\epsilon_{3,n}, \ldots, \epsilon_{r+2,n}$ be $r$ random variables such that $\max_i |\epsilon_{i,n}| = O_p(n^{-1/2})$. Then the solution $(\tilde{x}_1, \ldots, \tilde{x}_r)$ to the following system of equations*

$$\sum_{k=1}^{r}\tilde{x}_k^a = \sum_{k=1}^{r} x_k^a + \epsilon_{a,n} \quad \text{for } 3 \leq a \leq r+2 \tag{37}$$

*satisfies*

$$\max_i |\tilde{x}_i - x_i| = O_p(n^{-1/2}).$$

*Proof.* Let $\Delta_i = \tilde{x}_i - x_i$ for $1 \leq i \leq r$. By the implicit function theorem, the system of equations (37) has a unique solution with probability tending to 1. Moreover, by the continuous mapping theorem, we have $\Delta_i = o_p(1)$. By the definition of $\epsilon_{i,n}$, for any $\varepsilon > 0$, there exist finite constants $M$ and $N$ such that

$$\mathbb{P}\left(\max_i |\sqrt{n}\epsilon_i| > M\right) < \varepsilon \quad \text{for all } n > N.$$

Therefore, it suffices to show that

$$\mathbb{P}\left(\max_i |\Delta_i| \leq C\max_i |\epsilon_{i,n}|\right) \to 1 \tag{38}$$

for some constant $C > 0$.

Note that

$$\sum_{k=1}^{r} \tilde{x}_k^a - \sum_{k=1}^{r} x_k^a = \sum_{k=1}^{r} (x_k + \Delta_k)^a - \sum_{k=1}^{r} x_k^a$$

$$= \sum_{k=1}^{r} a x_k^{a-1} \Delta_k + O_p(\max_k \Delta_k^2).$$

We then calculate that

$$\sum_{k=1}^{r} a x_k^{a-1} \Delta_k = \tilde{\epsilon}_{a,n} \quad \text{for } 3 \le a \le r+2,$$

where $\tilde{\epsilon}_{a,n} = \delta_a + \epsilon_{a,n}$ and $\delta_a = O_p(\max_i \Delta_i^2)$.

For the above linear system of equations, by our assumption on $x_i$ (similar to the arguments in (29)), it has a unique solution of the form

$$\Delta_i = \sum_{j=3}^{r+2} a_{i,j} \tilde{\epsilon}_{j,n}, \tag{39}$$

where $a_{i,j}$ are constants depending only on $x_1, \ldots, x_r$. By combining (39) and the fact that $\max_a |\delta_a| = O_p(\max_i \Delta_i^2)$ and $\Delta_i = o_p(1)$, we conclude that (38) follows. $\qquad\square$

**Lemma L.6.** *Suppose that $x_1, \ldots, x_r$ are $r$ real numbers satisfying $|x_1| > |x_2| > \cdots > |x_r| > 0$, and that $\tilde{x}_1, \ldots, \tilde{x}_r$ are $r$ random variables satisfying $\max_i |\tilde{x}_i - x_i| = O_p(n^{-1/2})$. Let $y_1, \ldots, y_r$ be $r$ non-zero real numbers, and let $\epsilon_{1,n}, \ldots, \epsilon_{r,n}$ be $r$ random variables satisfying $\max_i |\epsilon_{i,n}| = O_p(n^{-1/2})$. Then the solution $(\tilde{y}_1, \ldots, \tilde{y}_r)$ to the following system of equations with respect to $(y_1, \ldots, y_r)$:*

$$y_a \ge 0, \quad \sum_{k=1}^{r} \tilde{x}_k^a \tilde{y}_k^2 = \sum_{k=1}^{r} x_k^a y_k^2 + \epsilon_{a,n} \quad \text{for } 1 \le a \le r, \tag{40}$$

*satisfies*

$$\max_i |\tilde{y}_i - y_i| = O_p(n^{-1/2}).$$

*Proof.* Note that

$$\tilde{x}_k^a \tilde{y}_k^2 - x_k^a y_k^2 = (\tilde{x}_k^a - x_k^a) y_k^2 + \tilde{x}_k^a (\tilde{y}_k^2 - y_k^2).$$

Since $\max_i |\tilde{x}_i - x_i| = O_p(n^{-1/2})$, we have $\max_i |\tilde{x}_i^a - x_i^a| = O_p(n^{-1/2})$. Therefore, equation (40) reduces to

$$y_a \ge 0, \quad \sum_{k=1}^{r} \tilde{x}_k^a (\tilde{y}_k^2 - y_k^2) = \tilde{\epsilon}_{a,n} \quad \text{for } 1 \le a \le r,$$

where $\max_a |\tilde{\epsilon}_{a,n}| = O_p(n^{-1/2})$. Moreover, since $\max_k |\tilde{x}_k^a| = O_p(1)$, and noting that the above system of equations is linear in $\tilde{y}_k^2 - y_k^2$ for $1 \le k \le r$, and that $r = O(1)$, we have

$$\max_k |\tilde{y}_k^2 - y_k^2| = O_p(n^{-1/2}).$$

Finally, recalling that $y_1, \ldots, y_r$ are non-zero, we conclude that

$$\max_k |\tilde{y}_k - y_k| = O_p(n^{-1/2}).$$

$\qquad\square$

**Lemma L.7.** *Under the assumptions of Theorem 3.6, we have*

$$\max_{1 \le i \le n} \max_{1 \le a \le r} \left| \frac{L_i^{(a)} - \mathbb{E}(L_i^{(a)} | U_i)}{\prod_{j=1}^{a} (n-j)} \right| = O_p\left( \sqrt{\frac{\log(n)}{n}} \right)$$

*where*

$$L_i^{(1)} = \sum_{i_1} E_{ii_1},$$

$$L_i^{(a)} = \sum_{i_1,\cdots,i_a \ distinct, i_k \neq i, 1 \leq k \leq a} E_{ii_1} \prod_{j=2}^{a} E_{i_{j-1}i_j} \ for \ a \geq 2.$$

*Proof.* We divide the proof into two steps. In **Step 1**, we show that

$$\frac{1}{\prod_{j=1}^{a}(n-j)} \max_i |L_i^{(a)} - S_{i,0}| = O_p\left(\sqrt{\frac{\log(n)}{n}}\right)$$

where

$$S_{i,0} = \sum_{i_1,\cdots,i_a \ distinct, i_k \neq i, 1 \leq k \leq a} f(U_i, U_{i_1}) \prod_{j=2}^{a} f(U_{i_{j-1}}, U_{i_j}).$$

In **Step 2**, we show that

$$\frac{1}{\prod_{j=1}^{a}(n-j)} \max_i |S_{i,0} - T_{i,1}| = O_p(n^{-1/2})$$

where

$$T_{i,1} = \mathbb{E}\left[\sum_{i_1,\cdots,i_a \ distinct, i_k \neq i, 1 \leq k \leq a} f(U_i, U_{i_1}) \prod_{j=2}^{a} f(U_{i_{j-1}}, U_{i_j}) \Big| U_i\right] = \mathbb{E}(L_i^{(a)}|U_i).$$

Then the proof is complete by combining the above two equations and noticing that $r$ is bounded.

**Step 1.** Let

$$S_{i,a-1} = \sum_{i_1,\cdots,i_a \ distinct, i_k \neq i, 1 \leq k \leq a} E_{ii_1} \prod_{j=2}^{a-1} E_{i_{j-1}i_j} f(U_{i_{a-1}}, U_i).$$

Then

$$\frac{1}{\prod_{j=1}^{a}(n-j)} \left(L_i^{(a)} - S_{i,a-1}\right) = \frac{1}{\prod_{j=1}^{a}(n-j)}$$

$$\sum_{i_1,\cdots,i_a \ distinct, i_k \neq i, 1 \leq k \leq a} E_{ii_1} \prod_{j=2}^{a-1} E_{i_{j-1}i_j} \left(E_{i_{a-1}i_a} - f(U_{i_{a-1}}, U_{i_a})\right).$$

$$(41)$$

Notice that $E_{i_{a-1}i_a} = I(U_{i_{a-1},i_a} \leq f(U_{i_{a-1}}, U_{i_a}))$ is binary, with $U_{i_{a-1},i_a} \sim \text{Uniform}(0,1)$ independently, and that $U_{ij}$ is independent of $U_k$ for any $i,j,k$. By Hoeffding's inequality in Theorem 2.6.2 of Vershynin [2018], we have for any $t > 0$,

$$\mathbb{P}\left(\frac{1}{\sqrt{n-a}} \left|\sum_{i_a \neq i, i_1,\cdots,i_{a-1}} \left(E_{i_{a-1}i_a} - f(U_{i_{a-1}}, U_{i_a})\right)\right| \geq t \Big| U_1,\cdots,U_n\right) \leq 2\exp(-ct^2)$$

where $c > 0$ is an absolute constant. Then

$$\mathbb{P}\left(\frac{1}{\sqrt{n-a}} \left|\sum_{i_a \neq i, i_1,\cdots,i_{a-1}} \left(E_{i_{a-1}i_a} - f(U_{i_{a-1}}, U_{i_a})\right)\right| \geq t\right)$$

$$= \mathbb{E}\left(\mathbb{P}\left(\frac{1}{\sqrt{n-a}} \left|\sum_{i_a \neq i, i_1,\cdots,i_{a-1}} \left(E_{i_{a-1}i_a} - f(U_{i_{a-1}}, U_{i_a})\right)\right| \geq t \Big| U_1,\cdots,U_n\right)\right) \leq 2\exp(-ct^2).$$

As a result, $\frac{1}{\sqrt{n-a}}\left|\sum_{i_a \neq i, i_1, \cdots, i_{a-1}}\left(E_{i_{a-1}i_a} - f(U_{i_{a-1}}, U_{i_a})\right)\right|$ are sub-Gaussian random variables, and we have

$$\mathbb{E}\max_{i_{a-1}}\left|\sum_{i_a \neq i, i_1, \cdots, i_{a-1}} E_{i_{a-1}i_a} - f(U_{i_{a-1}}, U_{i_a})\right|/(n-a) = O\left(\sqrt{\frac{\log(n)}{n}}\right).$$

By recalling (41) and the fact that $E_{ij}$'s are binary, we have

$$\frac{1}{\prod_{j=1}^{a}(n-j)}\mathbb{E}\max_i|L_i^{(a)} - S_{i,a-1}| = O\left(\sqrt{\frac{\log(n)}{n}}\right).$$

Similarly, let

$$S_{i,a-2} = \sum_{i_1, \cdots, i_a \text{ distinct}, i_k \neq i, 1 \leq k \leq a} E_{ii_1}\prod_{j=2}^{a-2} E_{i_{j-1}i_j} f(U_{i_{a-2}}, U_{i_{a-1}})f(U_{i_{a-1}}, U_{i_a}).$$

Then

$$\frac{1}{\prod_{j=1}^{a}(n-j)}\left(S_{i,a-1} - S_{i,a-2}\right) = \frac{1}{\prod_{j=1}^{a}(n-j)}\sum_{i_1, \cdots, i_a \text{ distinct}, i_k \neq i, 1 \leq k \leq a} E_{ii_1}$$

$$\times \prod_{j=2}^{a-2} E_{i_{j-1}i_j}\left(E_{i_{a-2}i_{a-1}} - f(U_{i_{a-2}}, U_{i_{a-1}})\right)f(U_{i_{a-1}}, U_{i_a})$$

$$= \frac{1}{\prod_{j=1}^{a-1}(n-j)}\sum_{i_1, \cdots, i_{a-1} \text{ distinct}, i_k \neq i, 1 \leq k \leq a-1} E_{ii_1}\prod_{j=2}^{a-2} E_{i_{j-1}i_j}\left(E_{i_{a-2}i_{a-1}} - f(U_{i_{a-2}}, U_{i_{a-1}})\right).$$

$$(42)$$

The same reasoning holds by Hoeffding's inequality to show that

$$\frac{1}{\prod_{j=1}^{a}(n-j)}\max_i|L_i^{(a)} - S_{i,a-2}| = O_p\left(\sqrt{\frac{\log(n)}{n}}\right).$$

Continuing this process iteratively and combining with the fact that $r$ is bounded, we obtain

$$\frac{1}{\prod_{j=1}^{a}(n-j)}\max_i|L_i^{(a)} - S_{i,0}| = O_p\left(\sqrt{\frac{\log(n)}{n}}\right).$$

$$(43)$$

**Step 2.** Let

$$T_{i,a-1} = \sum_{i_1, \ldots, i_a \text{ distinct}, i_k \neq i, 1 \leq k \leq a} f(U_i, U_{i_1})\prod_{j=2}^{a-1} f(U_{i_{j-1}}, U_{i_j})\mathbb{E}\left(f(U_{i_{a-1}}, U_{i_a}) \mid U_{i_{a-1}}\right).$$

Then, we have

$$\max_i \frac{1}{\prod_{j=1}^{a}(n-j)}|S_{i,0} - T_{i,a-1}|$$

$$= \max_i \frac{1}{\prod_{j=1}^{a}(n-j)}\left|\sum_{i_1, \ldots, i_a \text{ distinct}, i_k \neq i, 1 \leq k \leq a} f(U_i, U_{i_1})\prod_{j=2}^{a-1} f(U_{i_{j-1}}, U_{i_j})\right.$$

$$\times\left.\left(f(U_{i_{a-1}}, U_{i_a}) - \mathbb{E}(f(U_{i_{a-1}}, U_{i_a}) \mid U_{i_{a-1}}))\right)\right|$$

$$= \max_i \frac{1}{\prod_{j=1}^{a-1}(n-j)}\left|\sum_{i_1, \ldots, i_{a-1} \text{ distinct}, i_k \neq i, 1 \leq k \leq a-1} f(U_i, U_{i_1})\prod_{j=2}^{a-1} f(U_{i_{j-1}}, U_{i_j})\right.$$

$$\times \frac{1}{n-a}\sum_{i_a}\sum_{k=1}^{r} \lambda_k G_k(U_{i_{a-1}})\left[G_k(U_{i_a}) - \int_0^1 G_k(u)\,du\right]\right|$$

$$\leq \frac{1}{n-a}\sum_{k=1}^{r}\left|\lambda_k M \sum_{i_a}\left[G_k(U_{i_a}) - \int_0^1 G_k(u)\,du\right]\right| = O_p(n^{-1/2}),$$

where we use the facts that $f(x, y)$ are bounded by 1, $G_k$ are bounded by $M$, and $U_i$ are i.i.d. random variables.

Similarly, let

$$T_{i,a-2} = \sum_{i_1,\ldots,i_a \text{ distinct},i_k \neq i,1\leq k\leq a} f(U_i, U_{i_1}) \prod_{j=2}^{a-2} f(U_{i_{j-1}}, U_{i_j}) \mathbb{E}\left( f(U_{i_{a-2}}, U_{i_{a-1}}) f(U_{i_{a-1}}, U_{i_a}) \mid U_{i_{a-2}} \right).$$

Then, we have

$$\max_i \frac{1}{\prod_{j=1}^a (n-j)} |T_{i,a-1} - T_{i,a-2}|$$

$$= \max_i \frac{1}{\prod_{j=1}^a (n-j)} \left| \sum_{i_1,\ldots,i_a \text{ distinct},i_k \neq i,1\leq k\leq a} f(U_i, U_{i_1}) \prod_{j=2}^{a-2} f(U_{i_{j-1}}, U_{i_j}) \right.$$

$$\left. \times \left( f(U_{i_{a-2}}, U_{i_{a-1}}) \mathbb{E}(f(U_{i_{a-1}}, U_{i_a}) \mid U_{i_{a-1}}) - \mathbb{E}\left( f(U_{i_{a-2}}, U_{i_{a-1}}) f(U_{i_{a-1}}, U_{i_a}) \mid U_{i_{a-2}} \right) \right) \right|$$

$$\lesssim \frac{1}{n} \left| \sum_{i_{a-1}} \sum_{k_1=1}^r \sum_{k_2=1}^r \lambda_{k_1} \lambda_{k_2} G_{k_1}(U_{i_{a-1}}) G_{k_2}(U_{i_{a-1}}) \int_0^1 G_{k_2}(u) \, du \right.$$

$$\left. - \sum_{i_{a-1}} \sum_{k=1}^r \lambda_k^2 \int_0^1 G_k(u) \, du \right|$$

$$\lesssim \frac{1}{n} \sum_{k=1}^r \left| \sum_{i_{a-1}} \left( G_k^2(U_{i_{a-1}}) - 1 \right) \right| + \frac{1}{n} \sum_{k_1 \neq k_2} \left| \sum_{i_{a-1}} G_{k_1}(U_{i_{a-1}}) G_{k_2}(U_{i_{a-1}}) \right| + O\left(\frac{1}{n}\right)$$

$$= O_p(n^{-1/2}),$$

where we use the facts that $f(x, y)$ are bounded by 1, $G_k$ are bounded by $M$, $r$ is bounded, $\int_0^1 G_k^2(u) \, du = 1$, $\int_0^1 G_i(u) G_j(u) \, du = 0$ for $i \neq j$, and that $U_i$ are i.i.d. random variables.

Similar arguments can be performed for $T_{i,a-3}, \ldots, T_{i,1}$. Since $a \leq r$ is bounded, by combining all the results, we obtain

$$\frac{1}{\prod_{j=1}^a (n-j)} \max_i |S_{i,0} - T_{i,1}| = O_p(n^{-1/2}), \tag{44}$$

where

$$T_{i,1} = \mathbb{E}\left[ \sum_{i_1,\ldots,i_a \text{ distinct},i_k \neq i,1\leq k\leq a} f(U_i, U_{i_1}) \prod_{j=2}^a f(U_{i_{j-1}}, U_{i_j}) \mid U_i \right].$$

Finally, the proof is complete by combining the results from equations (43), (44), and noting that $r$ is bounded. $\square$

