# OpenReview forum: "Low-Rank Graphon Learning for Networks"
_NeurIPS.cc/2025/Conference — NeurIPS 2025 poster_

### Official Review · Reviewer_SWT9 · 2025-06-21

**Clarity:** 3
**Significance:** 2
**Originality:** 3
**Rating:** 4
**Confidence:** 4

**Summary:**

The paper addresses the problem of estimating the graphon function  $f: [0,1]^2 \to [0,1]$ that governs the edge probabilities in an undirected random graph. Each of the $n$ nodes in the graph is associated with an unobserved latent variable $U_i$, drawn independently from the uniform distribution on $[0,1]$. Conditional on these latent variables, the presence of an edge between nodes $i$ and $j$ is determined by a Bernoulli trial with success probability $f(U_i, U_j)$.

The authors propose new algorithms for estimating the graphon $f$, and establish that, under certain structural assumptions, the estimator $\hat{f}$ achieves a convergence rate of  $\sqrt{\frac{\log(n)}{n}}$ in the supremum norm. A key assumption is that the true graphon $f$ admits a decomposition as a finite sum of $r$ orthogonal, rank-one symmetric functions, i.e., functions of the form $ G(u)G(v)$, where each $G$ is a univariate function. Furthermore, it is assumed that the functions $G$ are Lipschitz continuous and that one of them is known to be monotone increasing. These structural constraints play a central role in enabling consistent and computationally efficient estimation.

**Questions:**

1. Could you provide some hints on why the conditions required in Section 3 could be acceptable in practice?

2. How should the equations (5) be solved?

3. You claim that your proposal based on counting subgraphs is computationally efficient. I did not find the justification of this claim in the main body. Did I miss something? If not, and if it is a crucial property of your method as compared to past work, it might be beneficial to move it to the main body of the paper.

4. Is it easy to extend the results of Section 3.2 to the case where $r$ might depend on $n$ and tend to infinity when $n$ tends to infinity? If yes, how then does $r$ enter into the rate?

**Ethical Concerns:**

["NO or VERY MINOR ethics concerns only"]

**Final Justification:**

I agree with the authors’ responses to my concerns regarding the orthogonality assumption and the condition that \( G_1 \) is monotone. However, I encourage the authors to consider these conditions jointly, together with the Lipschitz condition.

I am fully satisfied with items 3-5 of their response. As for item 6, I find the answer less convincing. Indeed, Algorithm 3 specifies that line 2 of Algorithm 2 should be used, which implies that the nonlinear system (5) must be solved. The computational complexity of this step (and therefore of Algorithm 3) remains unclear.

Finally, the fact that only the case of fixed \( r \) is considered remains a weakness of the paper.

Overall, I believe the paper merits publication. However, given the weaknesses noted in the reviews (some of which were not fully addressed in the rebuttal), I consider “weak accept” to be the appropriate score.

**Limitations:**

yes

**Quality:**

3

**Strengths And Weaknesses:**

**Strengths**
1. I enjoyed reading this paper. It is well polished and pleasant to read.
2. All the math details that I checked appear to be correct.
3. The obtained rates of estimation match the best known rates in the problem of graphon estimation.

**Weaknesses**
1. The conditions imposed on the true graphon appear to be quite strong, especially when it comes to considering the case $r > 1$. Indeed, both the condition that the $G_i$s are orthogonal and that they are orthogonal to the constant function is a strong assumption that has no reason to be satisfied.
2. The condition 3.7 that essentially requires that $G_1$ is an increasing function is overly restrictive. I cannot think of a relevant example of application in which such a condition could be acceptable.
3. The paper would be much better if more intuitions were given about the construction of the estimators. Here are some examples of intuitive explanations that can be added, but of course this is just a sample:
   a. The estimator provided at line 150 is based on the fact that the $i$th order statistic of the sample of uniform in $[0,1]$ random variables is close to $i/(n+1)$.
   b. The rate of convergence $n^{-1/2}$ corresponds to the standard nonparametric rate $N^{-\beta/(2\beta + d)}$, where $\beta$ is the smoothness, $d$ is the dimension and $N$ is the sample size. Since in the problem under consideration the observations are the edges, the sample size $N$ is $n^2$. The Lipschitz assumption implies that $\beta = 1$, and the dimension is obviously $d = 2$.

4. Many relevant references on graphon estimation are missing. I am not claiming that these references have a direct connection to the obtained results, but they have to be discussed in order to situate the current work in the landscape of already existing results. The papers I have in mind are:
- E. Donier-Meroz, A. Dalalyan, F. Kramarz, P. Choné, X. D'Haultfoeuille (2023). *Graphon Estimation in bipartite graphs with observable edge labels and unobservable node labels*. arXiv:2304.03590
- O. Klopp and N. Verzelen. *Optimal graphon estimation in cut distance*. *Probability Theory and Related Fields*, 174(3), pp. 1033–1090 (2019)
- O. Klopp, A. Tsybakov and N. Verzelen. *Oracle inequalities for network models and sparse graphon estimation*. *Annals of Statistics*, 45(1), pp. 316–354 (2017)

---

> ### Author Rebuttal · Authors · 2025-07-31
>
> Thank you for your review and the positive feedback. We truly appreciate your recognition of our writing, your checking of our mathematics, and your affirmation of our rate. We will address each of your comments one by one in the following response.
>
> **1. Regarding the conditions imposed on the graphon.**
>
> Thank you for your comment. We would like to clarify that the orthogonality assumptions are standard in the literature for functional decompositions and low-rank models. Formally, the assumption is a natural consequence of the Hilbert-Schmidt theorem, which guarantees that any symmetric, square-integrable kernel admits an eigen-decomposition with orthogonal eigenfunctions (see e.g., Chapter 10 in [4]). For the relationship between graphons and the Hilbert-Schmidt norm, we refer to Section 2.3 of [5]. Therefore, these conditions do not limit the generality of our approach, but rather allow us to present the results and the estimation procedure in a more interpretable and mathematically tractable way.
>
> **2. Regarding condition 3.7.**
>
> Thank you for your insightful question. In graphon estimation, it is well-known that the graphon is geneally non-identifiable - it is only defined up to measure-preserving transformations (see, for example, [1] for discussions on identifiability). Therefore, when estimating the graphon itself, the literature generally requires additional conditions. One example is [3], where the authors assume that the true graphon satisfies certain Lipschitz conditions and has sparse gradients. Our condition 3.7 serves a similar purpose, enabling the estimation of the graphon $f$.
>
> Regarding estimating graphons with reference to $G_1$, we can relax our approach by considering each $G_i$ ($i = 1, \\ldots, n$) as a potential reference function. For each $G_i$, we estimate the graphon $f$ as described in our method referencing $G_1$, then compare the expected motif (e.g. triangles, stars, and so on) densities from the estimated $\hat{f}$ with the observed motif densities in the data. By evaluating this criterion for all $i$, we select the $\\hat{f}$ that best matches the empirical motif distributions. This relaxation yields a more flexible approach, and we will include a remark in the paper to clarify this improvement. We leave the exploration of weaker conditions as future work.
>
> **3. Regarding intuitions about the construction of the estimators.**
>
> Thank you for your detailed and helpful suggestions. We especially appreciate your examples clarifying the intuition behind the estimator based on order statistics and the convergence rate in terms of smoothness, dimension, and sample size. We will incorporate these explanations into the revised manuscript to improve clarity and help readers better understand our methods. Thank you again for your valuable feedback.
>
> **4. Regarding the literature review.**
>
> Thank you for pointing out these important references. Discussing a broader range of related work helps position our contributions more clearly. In the revised version, we will cite and discuss relevant works on graphon estimation, to provide a more complete literature review.
>
> **5. Regarding the solution of equations (5).**
>
> Thanks for your question. In our implementation, we solve it numerically using the nleqslv package in R, which is designed for finding roots of nonlinear systems. This package applies standard numerical methods (such as Broyden’s or Newton-type algorithms) to obtain solutions efficiently. The speed of solving the equation is fast in practice, and we have not observed any anomalies. We will mention this in the revision.
>
> **6. Regarding computational efficiency of our algorithm.**
>
> Thank you for your comment. We believe Appendix A.1 addresses your concern. In Appendix A.1, Algorithm 3 shows how lines and cycles allowing repeated nodes can be efficiently computed via matrix multiplication, serving as a surrogate for counting lines and cycles without repeated nodes. This approach is further justified by Lemma A.2 and Theorem A.3, which demonstrate that the approximation error introduced is small and does not affect the error rate in our main theorem. Due to space constraints, these details were omitted from the main text but will be incorporated in the revised version.
>
> **7. Regarding the case where $r$ diverges $n$.**
>
> Thank you for your question. We treat $r$ as fixed in this paper, and we believe that the theory for the regime where $r$ grows with $n$ is fundamentally different. If $r$ grows with $n$, the model complexity increases alongside network size, making inference significantly more challenging - similar to high-dimensional or nonparametric settings. Consequently, theoretical guarantees and methods for fixed-rank cases generally do not apply, and larger sample sizes are needed for comparable estimation accuracy. Addressing this growing-rank regime requires new theoretical tools and remains an open research problem. Therefore, we leave it for future work.
>
> Empirically speaking, when $r$ increases, a larger sample size is required to achieve the similar performance (e.g., in MSE and sup-norm), which aligns with our simulation experiments.
>
> [1] Lovász, László. Large networks and graph limits. Vol. 60. American Mathematical Soc., 2012.
>
> [2] Olhede, Sofia C., and Patrick J. Wolfe. "Network histograms and universality of blockmodel approximation." Proceedings of the National Academy of Sciences 111, no. 41 (2014): 14722-14727.
>
> [3] Chan, Stanley, and Edoardo Airoldi. "A consistent histogram estimator for exchangeable graph models." In International Conference on Machine Learning, pp. 208-216. PMLR, 2014.
>
> [4] Ash, Robert B. Real analysis and probability: probability and mathematical statistics: a series of monographs and textbooks. Academic press, 2014.
>
> [5] Xu, Jiaming. "Rates of convergence of spectral methods for graphon estimation." International Conference on Machine Learning. PMLR, 2018.

---

> > ### Comment · Reviewer_SWT9 · 2025-08-04
> > **Feedback to the authors' response**
> >
> > I would like to thank the authors for their answers. I will maintain my score.

---

### Official Review · Reviewer_BVhG · 2025-06-29

**Clarity:** 2
**Significance:** 1
**Originality:** 2
**Rating:** 4
**Confidence:** 3

**Summary:**

This paper offers a framework for estimating the connectivity matrix and the underlying function of a low-rank graphon through subgraph counts (AKA motifs). The authors' main theorem is an error guarantee in the sup-norm for both the connectivity matrix $P$ (Theorem 3.6) and the graphon function $f$ (Theorem 3.8). The former is noted to match pre-existing minimax error rates for smooth graphon estimation. Additionally, the authors perform simulation studies to compare their method with various prior algorithms, in terms of mean-squared errors and runtimes.

**Questions:**

- Literature review seems sparse: can you cite more papers on matrix sensing / matrix completion, as well as Holder-smooth graphon estimation? Putting in an appendix would be okay.

Chen, Yudong, et al. "Completing any low-rank matrix, provably." The Journal of Machine Learning Research 16.1 (2015): 2999-3034.

KLOPP, OLGA, ALEXANDRE B. TSYBAKOV, and NICOLAS VERZELEN. "ORACLE INEQUALITIES FOR NETWORK MODELS AND SPARSE GRAPHON ESTIMATION." The Annals of Statistics 45.1 (2017): 316-354.

Jalan, Akhil, et al. "Transfer Learning for Latent Variable Network Models." The Thirty-eighth Annual Conference on Neural Information Processing Systems.

- Eq. 1 seems to imply that the graphon is “almost” Lipschitz, since each G_k is L^2, so it must be bounded outside a set of measure zero in [0,1]. So, I understand the main difference between your model and smooth graphons to be this measure zero set.

How critical is this set of measure zero for
(a) real-world modeling applications such as the biological networks you cite, and

(b) theoretical work such as your estimation algorithms, as compared to USVT or Gao’s averaging technique?

Some discussion on both points would be helpful.

- The discussion of theoretical contributions on line 42 begins with Lemmas A8-A10. Could these be consolidated into a single statement and put in the main body? If they are a core contribution of the work, I feel they should be at least discussed in the main text, even if space constraints prevent the proof from being included therein.

- Line 126: Can Xu 2018 not also take advantage of the rank-1 structure to do better than O(n^3) time? If I recall, Xu’s algorithm is bottlenecked by computing the SVD itself.

- Theorem 3.1: You are citing previous minimax lower bounds later for smooth graphon estimation, so I understand these are a special case of your model. What is an example of a graphon that falls into your model but does not meet the piecewise Holder assumption?

- Theorem 3.2: The use of a canonical G_1 seems similar to the idea of estimating graphons up to equivalence relations as defined in e.g. Klopp 2017, among others. Can you explain the differences?

- Line 187: What is the citation is needed for the complexity of matrix multiplication? I think would be more readable if you just wrote $O(n^\omega)$.

- Line 193-194: You emphasize that your method does not require smoothness assumptions, but Theorem 3.2 requires a Lipschitz G_1, which would imply a Lipschitz f in the rank-1 case. Lipschitz smoothness is in fact stronger than Holder smoothness as in Xu 2018. Can you clarify how your method does not require smoothness?

- Assumption 3.5(iii): Similarly, the assumption of bounded G_k functions, along with them being in L^2, seems to imply a lot of regularity, since the G_k must be Lipschitz. How precisely does your model differ? I apologize for being repetitive.

- Table 1 would benefit from bolds / italics to emphasize the best performing methods in each scenario

- Table 1: Is the runtime an average across all 100 runs? Reporting stdev or quantile interval of runtimes would be helpful.

- Table 1: It seems USVT is comparable in MSE across all scenarios except #4; however, even here it is comparable in terms of max error. Why is USVT so effective in all of these examples? Are there any motivating examples of graphons where USVT is clearly worse?

- Table 2: Why does sparsity make your method so much better than others? Discussion would be helpful.

- Table 2: It would be helpful to see how methods’ performance scales with sparsity, ranging from $\rho = 1/\sqrt{n}$ up to $\rho = 1$.

- Appendix A6: Can you report the MSE numbers and comparisons to your baseline methods (USVT, neighborhood smoothing, etc) for these experiments? With the figures as they are now, it is very unclear why your method should be preferable for these real-world graphs.

**Ethical Concerns:**

["NO or VERY MINOR ethics concerns only"]

**Final Justification:**

In light of the additional discussion in the review period, especially the new experiments which examine a range of sparsity parameters for $\rho$, and the responses to Q6 and Q7 for reviewer n6Ak, I will update my score to a 4.

**Limitations:**

There is not much discussion of the limitations of their work, aside from a suggestion on future work in the conclusion. I would suggest a deeper discussion of:

- The unique aspects of their model as opposed to smooth graphon models.

- The real-world applicability of their model, perhaps to the graphons they study in Appendix A.6 or others.

- A more extensive discussion of why their method only seems to outperform baselines in a sparse graphon regime.

**Paper Formatting Concerns:**

No formatting concerns.

**Quality:**

2

**Strengths And Weaknesses:**

The paper's main strength is presenting and addressing graphon estimation in a new framework: namely, estimating both the graphon connectivity matrix and the underlying function, through efficient algorithms that leverage motifs. Additionally, the numerical experiments are extensive, and demonstrate their method is advantageous (at least in a very low sparsity regime of $\rho = 1/\sqrt{n}$). The algorithm is clean, scalable, and intuitive.

The main weaknesses are:

1. Unclear scope for their model of low-rank graphons, which seem quite similar to the well-studied case of piecewise Holder-smooth graphons. Experiments show that prior methods are mostly competitive even for the authors' chosen simulated graphons (see my questions below), so it is not clear what sort of graphon problems really fit their specific theoretical assumptions and algorithm.

2. Lack of real-world experiments, which appear in Appendix A.6 but in very limited form (see my questions) below.

---

> ### Author Rebuttal · Authors · 2025-07-31
>
> Thank you for taking the time and effort to review our work, and for your positive feedback on our algorithm and theory. We will now address each of your questions one by one.
>
> **1. Regarding additional literature review.**
>
> Thank you for your helpful suggestions and for recommending these excellent papers. We agree that a more comprehensive literature review will strengthen the manuscript. We will cite these works and provide an expanded discussion on matrix sensing, matrix completion, and Hölder-smooth graphon estimation in the revised version, including additional details in the appendix as suggested.
>
> **2. Regarding Eq. (1) and the "almost" Lipschitz property of the graphon.**
>
> Equation (1) states that our graphon $f$ is low-rank, with each component $G_k$ in $L_2$. We address your concerns from two angles: the low-rank property and the measure-zero set.
>
> From the low-rank perspective, this is a key distinction from approaches like USVT. The low-rank assumption provides two main benefits: (i) it enables consistent estimation of both the connection probability matrix $P$ and the graphon $f$ (see our response to Reviewer 1DRW for more on estimating $f$); and (ii) it improves computational efficiency, with complexity comparable to matrix multiplication. In contrast, methods like those in [4], [8] incur much higher computational costs.
>
> Regarding the measure-zero set and continuity assumptions, unlike Gao [5] and USVT [2], we do not require the graphon $f$ to be continuous or in stochastic block model (SBM) form for estimating $P$. This flexibility is practical, as real-world networks, such as biological networks, can exhibit sharp structural heterogeneity or abrupt changes that violate smoothness assumptions. Allowing for discontinuities on measure-zero sets makes our model more flexible and realistic.
>
> Such irregularities challenge classical methods like the Network Histogram [4] and SAS [6], which may fail or yield biased estimates. Moreover, Gao’s averaging technique has a relatively high computational complexity.
>
> While our approach offers these theoretical and practical advantages, further investigation is needed to fully assess the impact of these assumptions on real-world datasets. We see this as an important avenue for future research.
>
> **3. Regarding Lemmas A8-A10.**
>
> Thank you for your suggestion. Lemmas A8–A10 are technical lemmas that primarily serve as auxiliary results for the main theoretical proofs. We will consolidate them into a single statement and incorporate the content into the main article.
>
> **4. Regarding Xu's method in rank 1 conditions.**
>
> If it is known a priori that the connection probability matrix is rank-1, a truncated version of Xu’s USVT method could, in principle, be accelerated to $O(n^2)$ time by computing only the top singular value and its corresponding singular vectors using efficient algorithms such as the power iteration method.   For the case of $r>1$, since they use SVD, their method appears to be slower than ours based on the simulations.
>
> **5. Regarding Theorem 3.1.**
>
> Thank you for your question. Theorem 3.1 concerns the estimation of the connection probability matrix $P$ in the rank $r=1$ setting. In this case, no additional smoothness or continuity conditions are required on $G_1$. It suffices that $G_1$ belongs to $L_2$ and is non-zero on a set of positive measure. Notably, $G_1$ may contain many discontinuities and does not need to be piecewise Lipschitz.
>
> **6. Regarding the use of a canonical $G_1$.**
>
> Thank you for your question. In Klopp et al. (2017), the identifiability issue is addressed by defining a distance between functions that accounts for measure-preserving transformations (see their Eq. (14) for details). In contrast, our approach resolves identifiability by defining a canonical graphon function.
>
> **7. Regarding the complexity of matrix multiplication.**
>
> Thank you for your suggestion. We will use the notation $O(n^\\omega)$ to clarify the expression. The relevant citation is [7], which we have now added to Appendix A.1 and will include in the main text.
>
> **8. Regarding the smoothness assumptions.**
>
> Thank you for your question. We would like to clarify that lines 193–194 refer to Remark 3.4, where we state that no smoothness assumptions are needed for estimating the connection probability matrix $P$. In contrast, Xu (2018) requires smoothness assumptions to estimate $P$.
>
> Additionally, our analysis focuses on the sup-norm error, whereas Xu (2018) primarily considers the $L_2$ norm. Their approach does not estimate the graphon function $f$ itself. The Lipschitz condition arises only when estimating $f$, which is a fundamentally more challenging and ill-posed problem due to identifiability issues.
>
> Assumption 3.5(iii) pertains to graphon estimation, which Xu (2018) cannot handle. The condition that $G_k$ is bounded can be relaxed, and we leave this for future work.
>
> **9. Regarding results in Table 1.**
>
> We will use boldface  to highlight the best-performing methods in the revised version. As for runtime values, they are averaged over 100 runs, and we will add the standard deviation (which is relatively small) in the revised manuscript.
>
> The rate of USVT is indeed nearly minimax optimal in terms of MSE under certain conditions [2], so it is not surprising that it performs so well. However, we emphasize that USVT cannot estimate the graphon function itself, which is the key distinction between our approach and theirs.
>
> **10. Regarding results in Table 2.**
>
> While our simulations show that our method works for sparse graphs, the theoretical guarantees for this regime are left for future work, as mentioned in the discussion. Intuitively, Equation (5) is homogeneous with respect to sparsity, suggesting that our theoretical results may extend to the sparse regime. A rigorous proof, however, would require extending the theoretical arguments in our paper.
>
> Regarding the performance of all methods as sparsity varies from $n^{-1/2}$ to $1$, we conducted additional simulations, and the results are shown in Table R1. As sparsity increases, the estimation error for all methods also increases. However, our method consistently outperforms others in the sparse regime and remains competitive as sparsity increases, highlighting its particular strength for estimating sparse networks.
>
> **Table R1. Results for estimating the connection probability matrix $P$ generated by sparse graphons across 100 independent trials.**
>
> |Graphon ID|Sparsity Parameter $\rho_n$|Method|MSE $(\times 10^{-4})$|Std. dev of MSE $(\times 10^{-6})$|Max. Error $(\\times 10^{-2})$|Std. dev of Max. Error $(\times 10^{-3})$|
> |----------|---------------------------|------|----------------------|----------------------------------|----------------------------|-----------------------------------------|
> |4|$n^{-1/3}$|Ours|0.178|0.663|5.722|9.922|
> | | |N.S.|31.285|140.98|99.907|0.15|
> | | |USVT|0.389|1.964|6.163|24.638|
> | | |Nethist|1.193|7.809|7.815|0.278|
> | | |SAS|0.597|3.787|99.72|0.985|
> | |$n^{-1/6}$|Ours|0.752|3.501|10.022|12.695|
> | | |N.S.|8.833|86.337|98.761|0.416|
> | | |USVT|1.348|6.356|12.44|11.336|
> | | |Nethist|13.66|89.165|27.699|1.149|
> | | |SAS|5.356|32.228|99.597|2.321|
> | |$n^{-1/10}$|Ours|1.116|4.874|12.412|14.103|
> | | |N.S.|3.264|10.317|38.125|129.087|
> | | |USVT|2.161|10.221|15.706|14.102|
> | | |Nethist|34.658|221.149|45.841|2.907|
> | | |SAS|14.681|59.42|93.407|72.598|
> |5|$n^{-1/3}$|Ours|0.258|1.205|4.93|7.207|
> | | |N.S.|29.992|170.322|99.969|0.167|
> | | |USVT|0.9|3.584|6.231|5.969|
> | | |Nethist|3.316|10.137|6.766|0.074|
> | | |SAS|0.821|4.783|97.805|101.723|
> | |$n^{-1/6}$|Ours|1.145|34.613|11.757|40.912|
> | | |N.S.|3.624|6.686|20.814|98.466|
> | | |USVT|3.039|11.992|12.511|9.675|
> | | |Nethist|30.673|72.197|23.993|0.395|
> | | |SAS|9.165|18.182|74.31|132.569|
> | |$n^{-1/10}$|Ours|1.338|33.349|10.409|30.081|
> | | |N.S.|4.852|10.989|16.784|9.73|
> | | |USVT|4.626|18.243|15.865|13.431|
> | | |Nethist|58.583|185.89|39.152|5.418|
> | | |SAS|9.281|21.39|93.009|68.993|
>
> **11.  Regarding quantitative graphon-estimation errors, and validating the quality of the real-data estimates**
>
> Thank you for your question. We kindly refer you to our response Q6 and Q7 to Reviewer n6Ak.
>
> **W1. Regarding the low-rank graphon setting, performance of the experiments, and the theoretical assumptions.**
>
> We kindly refer you to our responses 2, 5, 6, 8, 9, 10.
>
> **W2. Regarding the real-world experiments.**
>
> We kindly refer you to our response Q7 to Reviewer n6Ak, where we conducted experiments to validate the quality of the real-data estimations.
>
> [1] Klopp et al., "Oracle inequalities for network models..." (2017): 316-354.
>
> [2] Xu, "Rates of convergence of spectral methods..." (2018): 5433-5442.
>
> [3] Borgs et al., "Graph limits and parameter testing..." (2006): 261-270.
>
> [4] Olhede & Wolfe, "Network histograms and universality..." (2014): 14722-14727.
>
> [5] Gao et al., "Rate-optimal graphon estimation..." (2015): 2624-2652.
>
> [6] Chan & Airoldi, "A consistent histogram estimator..." (2014): 208-216.
>
> [7] Vassilevska Williams, "Multiplying matrices faster..." (2012): 887-898.
>
> [8] Bickel et al., "The method of moments and degree distributions..." (2011): 2280-2301.

---

> ### Comment · Area_Chair_nAtW · 2025-08-05
>
> Dear Reviewer BVhG,
>
> The authors appreciate your questions that raised in your comments, and they have carefully addressed them in their rebuttals. The author really expect to share views with you to improve their submission further. Could you find time to join the discussion before the end of the author-reviewer discussion period? Many thanks for your time!
>
> Bests,
> Your AC

---

> ### Comment · Reviewer_BVhG · 2025-08-05
>
> I thank the authors for their extensive answers to my questions. I have updated my score to a 4 to reflect my increased confidence in the results of the paper, especially the new experimental results that I highlighted. I trust that these will make it into the final version of the manuscript.
>
> Regarding literature review: I think the citation of these works will help situate the work, and agree with other reviewers regarding the value of citing major works in graphon estimation such as that of Klopp et al.
>
> Regarding #4: I think including this discussion in the remark quoted would be helpful. As it is, I don't believe that the computational complexity of each method makes a major difference, but since the discussion occurs in a subsection about rank-1 settings anyways, it should be properly compared.
>
> Finally, I would note that my confusion about the smoothness and regularity assumptions of the paper (Q2 and Q8), especially as compared to the Holder-smooth setting, might be shared by other readers, so I would ask that the authors include these remarks somewhere in the updated manuscript.

---

> > ### Author Response · Authors · 2025-08-06
> >
> > Thank you very much for your feedback and for updating your score. We greatly appreciate your suggestions regarding the literature review, computational complexity, and the smoothness assumptions. Regarding the computational complexity discussion in the rank-1 subsection, we have already included a comparison with the Power Iteration method in Table 1 of the original manuscript. However, we will provide additional discussion and ensure that this comparison is presented more clearly in the revised manuscript. We will ensure that the points you mentioned are incorporated into the revised manuscript. Thank you again for your valuable input.

---

### Official Review · Reviewer_n6Ak · 2025-07-02

**Clarity:** 2
**Significance:** 2
**Originality:** 3
**Rating:** 4
**Confidence:** 3

**Summary:**

The authors proposed a methodology that estimates both the connection probability matrix and its underlying graphon at low computational cost. The approach leveraged the rank-r eigen-decomposition of a graphon into r eigenvalues and eigenfunctions. These quantities were recovered by counting cycles and paths of prescribed length, which leads to an estimate of the connection probability matrix. Identification issue was handled via sorting, and linear interpolation was used to extend the discrete estimates of the eigenfunctions to a continuous graphon.

**Questions:**

1. How can this work tangibly benefit the AI research community or How can this work be related to the scope of the conference?

2. Can you provide the missing results for IDs 1, 6, 7 and the runtime column in Table 2?

3. What happens if the target graphon is high-rank? Is the method scalable or accurate in that regime?

4. Is low-rank (less than or equal to 3) approximation applicable to real-world data?

5. Could you compare computational cost as a function of rank?

6. Could you report quantitative graphon-estimation errors (e.g., norm)?

7. How do you validate the quality of the real-data estimates?

**Ethical Concerns:**

["NO or VERY MINOR ethics concerns only"]

**Final Justification:**

My concerns about a relevance to this venue and a lack of experimental verification have now been resolved through rebuttals. However, a concern regarding readability, presentation, and reproducibility still remains. I hope that the authors can significantly improve the readability while surely providing a source code upon acceptance.

**Limitations:**

- Unclear practical impact
- Limited experimental verification

**Quality:**

2

**Strengths And Weaknesses:**

<Strengths>

1. Mathematically well-founded:
The methodology was supported by rigorous proofs; e.g., Theorems 3.2, 3.6, and 3.8 establish convergence rates.

2. Solid methodology:
To my knowledge, the authors introduced a new methodology to estimate both a connection probability matrix and graphon of a graph while remaining computationally efficient.

<Weaknesses>

1. Writing is hard to follow
- The word “learning” in the title is potentially misleading; the work concerns “estimation”, not machine learning.
- Key background concepts such as graphon, identification issue, connection probability matrix were introduced too briefly.
- No visual aids were provided to help readers grasp the pipeline.
- Several derivations, particularly those involving interpolation, omitted explanatory text between equations.
- Practical relevance was under-explained: concrete downstream tasks or benchmarks would help.

2. Insufficient experimental verification
- Table 1 and Table 2 were inconsistent: Table 2 omitted runtime results and the entries for IDs 1, 6, and 7.
- Experiments appeared limited to rank less than or equal to 3.
- Graphon estimation was assessed only qualitatively (Fig. 1); no quantitative error metric on graphon (not connection probability matrix) is reported.
- Real-data experiments were presented, but there is no evaluation of how accurate the estimated graphons or matrices are.

3. The source code is not available.
Thus, there is limitation in reproducing the proposal.

---

> ### Author Rebuttal · Authors · 2025-07-31
>
> Thank you for reviewing our paper. We appreciate your recognition of our proof, method novelty, and algorithm effectiveness. Below, we address your questions one by one.
>
> **1. Concerns about clarity, use of “learning” in the title, and lack of context or motivation.**
>
> Thank you for the feedback. We acknowledge that several aspects of the presentation can be improved to enhance clarity and accessibility.
>
> *(1) On "learning" usage:* Our use of "learning" follows its statistical meaning in latent structure estimation (e.g., [14], [15]). We understand it could imply a focus on predictive machine learning and are open to changing the title to "estimation" to avoid confusion.
>
> *(2) On key concepts:* We agree the introduction of graphons, identifiability, and the connection probability matrix was too brief. In the revision, we’ll expand this section, cite standard references [1], [2], [3], and include a toy example with $f(u,v) = \\cos(u-v)$ (see our first response to Reviewer 1DRW) for better intuition.
>
> *(3) On the lack of visual aids:* We will include a schematic figure to illustrate the overall modeling and estimation pipeline, which will help readers understand the structure and flow of our method.
>
> *(4) On missing exposition between equations:* We agree that several derivations, particularly in Section 3.2, can be made more accessible. We will revise this section to add explanatory text and clarify the interpolation-based estimation procedure (see also our first response to Reviewer 1DRW).
>
> *(5) On practical relevance and downstream tasks:* We appreciate this point. In the revision, we’ll add a subsection discussing the utility of estimating the graphon $f$ beyond recovering $P$, focusing on predicting motifs and higher-order structures (see our second response to Reviewer 1DRW). We’ll also clarify its role in network comparison and structure generalization.
>
> **2. Regarding insufficient experimental verification and evaluation.**
>
> Thank you for raising these important points. We address each concern as follows:
>
> *(1) Inconsistency between Tables 1 and 2:* We acknowledge that Table 2 omits runtime results and entries for graphon IDs 1, 6, and 7 to focus on key illustrative cases. We will clarify this in the revision to avoid confusion and include a more comprehensive summary of runtimes and results across all graphons.
>
> *(2) Experiments limited to rank $r\\leq 3$:* While our main experiments focus on low ranks for clarity and interpretability, our method extends naturally to higher ranks. Due to space constraints, we prioritized detailed analysis for $r\\leq 3$ but will include additional experiments for larger ranks in the supplementary material.
>
> *(3) Quantitative error metrics on graphon estimation:* We agree that quantitative evaluation of graphon estimates is crucial. In the revision, we will add such metrics, including integrated squared error and sup-norm errors, to complement the qualitative assessments in Figure 1.
>
> *(4) Real-data experiments and accuracy evaluation:* While real-data experiments primarily demonstrate practical applicability, we acknowledge the importance of assessing the accuracy of estimated graphons or connection probability matrices. In our response to Q7 below, we provide an illustration by evaluating the estimation quality through motif counting.
>
> **3. Regarding source code.**
>
> We fixed the random seed and ran 100 repetitions to ensure reproducibility. The code will be publicly available on GitHub, and the link will be included in the revision. All datasets used are publicly available and cited in the manuscript.
>
> **Q1: Relevance to AI.**
>
> Overall, statistical network analysis and graphon estimation are increasingly important in machine learning and AI, as reflected by recent work at NeurIPS and related conferences (e.g., [8], [9], [10]). Our methodology advances efficient and accurate network model estimation, with broad impact in domains such as social network analysis [4] and knowledge graphs [5].
>
> **Q2. Regarding the results for IDs 1, 6, 7 and the runtime column in Table 2.**
>
> Due to page limits, we omitted the results for graphon IDs 1, 6, and 7 under the sparse setting in Table 2, but our method still outperforms others in these cases. The runtime results in Table 2 are similar to those in Table 1, as the network size remains the same. Due to space constraints, we were unable to present these results, but we will include all of them in the revised manuscript.
>
> **Q3. Regarding the scalability of our method to high ranks.**
>
> Our theoretical results hold for any fixed rank $r$, including large values. However, in this paper, we treat $r$ as fixed and do not consider the regime where $r$ grows with the sample size $n$, which poses fundamentally different and more challenging problems. We leave this important direction for future work. Empirically, as $r$ increases, a larger sample size is required to maintain comparable performance (e.g., in terms of MSE), which is consistent with our simulation results. To further demonstrate scalability, we will include additional experiments with $r=5$ in the revised version.
>
>   **Q4. Regarding application of low-rank methods to real-world data.**
>
> While our low-rank setting assumes $r\ll n$, the rank $r$ can certainly be larger than $3$. Low-rank approximations are widely used in real-world data analysis. For example, the stochastic block model (SBM) has been extensively applied for community detection in social [11], biological [12], and geological networks [13] due to its interpretability and ability to capture community structures. As noted in lines 101–104 of our manuscript, our framework generalizes low-rank models like the random dot product graph (RDPG) and SBM.
>
> **Q5. Regarding the relationship between computational cost and rank.**
>
> When the rank $r = O(1)$ is fixed, the computational complexity of our method is $O(r n^\\omega)$, where $n^\omega$ denotes the complexity of matrix multiplication. Since $r$ is a constant, this simplifies to $O(n^\\omega)$, indicating that the cost is dominated by matrix operations.
>
> **Q6. Regarding quantitative graphon-estimation errors.**
>
> To provide further illustration, we conducted additional experiments on graphon IDs 3 and 4, comparing the estimated graphon with the true graphon using the maximum entrywise error (i.e., the sup-norm). Due to space constraints, results for other graphons are omitted here but will be included in the revised manuscript.
>
> The results, summarized in the table below, show that the sup-norm errors of the graphon estimates closely match those of the estimated connection probability matrices, as expected. This demonstrates that the estimation error is well controlled in practice, consistent with our theoretical guarantees in Theorems 3.2 and 3.8.
>
> *Table R2. Simulated sup norms for graphons across 100 independent trials.*
>
> | Graphon ID | Sup-norm ($\times 10^{-2}$) | Std. Dev. ($\times 10^{-3}$) |
> |-|-|-|
> |3|12.053|10.738|
> |4|15.539|12.037|
>
> **Q7. Regarding validating the quality of the real-data estimates.**
>
> Using the U.S. political blog dataset as an illustrative example, we demonstrate the advantages of our method through extensive comparisons with existing approaches.
>
> Specifically, we estimated the underlying graphon from the observed network and computed the expected densities of various motifs, such as triangles, squares, and 5-cycles (see also our response to Reviewer 1DRW for technical details). For competing methods, we generated synthetic networks from their estimated connection probability matrices and performed motif counting on these networks. We then compared the absolute differences in motif counts - normalized by the total possible in a complete graph of the same size - between the original and generated networks. The results, summarized in Table R3, show that our method consistently achieves smaller errors in motif prediction than competing approaches.
>
> *Table R3. Motif count errors and runtime for each method.*
>
> |Method|Triangle counting error$(\times 10^{-4})$|Run time(s)|Square counting error$(\times 10^{-5})$|Run time(s)|5-cycle counting error$(\times 10^{-6})$|Run time(s)|
> |-|-|-|-|-|-|-|
> |Ours|0.252|0.167|0.761|0.198|3.848|0.175|
> |N.S.|1.709|119.812|3.758|120.837|9.039|123.863|
> |Nethist|8.087|19.082|1.809|20.523|4.383|22.427|
> |USVT|4.449|15.97|1.712|17.098|5.149|19.439|
> |SAS|1.327|3.018|2.256|4.730|4.960|5.237|
>
> Moreover, as shown in Table R3, our approach significantly improves computational efficiency. After estimating the graphon, the expected density of any motif in networks of any size can be quickly approximated using the plug-in method, with only a constant number of matrix operations per sample. In contrast, existing methods require explicit motif counting in generated networks, which is computationally expensive (see [7] for details). This blend of efficiency and accuracy is particularly beneficial for downstream tasks where motif statistics are crucial (e.g., [6]).
>
> [1] Chan, S., & Airoldi, E. "Consistent histogram estimator..."
>
> [2] Zhou, Z., & Amini, A. A. "Spectral clustering for..."
>
> [3] Abbe, E., & Sandon, C. "Community detection in..."
>
> [4] Li, Y., et al. "FairLP: Fair link..."
>
> [5] Nickel, M., et al. "Review of relational..."
>
> [6] Milo, R., et al. "Network motifs: Building..."
>
> [7] Jin, J., et al. "Counting Cycles with..."
>
> [8] Li, M., et al. "Network change point..." NeurIPS, 2022.
>
> [9] Gaucher, S., & Klopp, O. "Optimality of variational..." NeurIPS, 2021.
>
> [10] Valdivia, E. A., & De Castro, Y. "Latent distance estimation..." NeurIPS, 2019.
>
> [11] Al-Anzi, B., et al. "Modeling modular structure..."
>
> [12] Holland, P. W., et al. "Stochastic blockmodels: First..."
>
> [13] Davari, A., et al. "Enhancing reservoir production..."
>
> [14] Aicher, C., et al. "Learning latent block..."
>
> [15] Kitson, N. K., et al. "Survey of Bayesian..."

---

> ### Comment · Reviewer_n6Ak · 2025-08-02
> **Further comments**
>
> Thank you for the detailed comments. Although the authors have devoted to addressing our concerns, some critical points still remain as follows:
>
> 1. Inconsistency between Tables 1 and 2:
> Could you provide the missing results now? It is uncommon to selectively omit results due to space limit.
>
> 2. Practical relevance and downstream tasks:
> Could you elaborate on the practical implications of your method and illustrate how it can be applied to specific downstream tasks now?
>
> 3. Request for additional experiments:
> In the appendix, you reported results for the case where the graph’s true rank is 1 but the estimated rank is 2. Could you also present experiments for the converse scenario, where the true rank is 2 and the estimated rank is 1?

---

> ### Author Response · Authors · 2025-08-02
>
> Thank you for taking the time to read our response and for raising additional questions. We will address your three concerns one by one.
>
> **Answer to Q1:**
> First, we would like to emphasize that all our simulations were conducted with fixed random seeds, ensuring full reproducibility. We will include the GitHub link to our code in the revised version of the paper.  As for the missing settings in Tables 1 and 2, we were unable to include them in the previous rebuttal due to space constraints. We now provide the complete results in Tables R4.
>
> **Table R4: Results for sparse graphons ID 1, 6, 7 characterized by $\rho_n = 1/\sqrt{n}$ across 100 repetitions.**
>
> |ID|Method|MSE($\times10^{-4}$)|Std.dev of MSE($\times10^{-6}$)|Max.error($\times10^{-2}$)|Std.dev of max.error($\times10^{-3}$)|Runtime(s)|Std.dev of runtime(s)|
> |--|------|--------------------|--------------------------------|--------------------------|-------------------------------------|----------|----------------------|
> |1|Ours|0.036|0.127|1.784|2.548|0.182|0.012|
> ||N.S.|19.224|45.848|99.665|0.000|113.929|0.876|
> ||Nethist|0.116|0.460|1.115|1.594|18.372|0.529|
> ||USVT|0.051|0.077|0.335|0.000|15.309|0.640|
> ||SAS|0.153|1.990|99.665|0.000|1.376|0.034|
> ||P.I.|0.249|0.735|1.500|0.000|0.392|0.039|
> |6|Ours|0.105|0.536|2.571|0.909|0.692|0.197|
> ||N.S.|42.530|69.027|99.984|0.099|112.049|1.937|
> ||Nethist|0.408|1.372|4.326|3.571|17.048|0.863|
> ||USVT|0.224|1.160|4.734|3.877|10.932|0.748|
> ||SAS|0.390|1.396|89.351|3.252|1.782|0.402|
> ||P.I.|0.415|2.055|2.821|4.047|0.983|0.502|
> |7|Ours|0.091|0.585|2.139|1.440|1.039|0.193|
> ||N.S.|29.434|103.240|99.858|1.310|124.024|1.849|
> ||Nethist|0.359|1.688|3.123|1.715|17.958|1.098|
> ||USVT|0.410|1.306|4.808|1.855|9.074|0.846|
> ||SAS|0.915|3.112|99.604|0.113|1.402|0.183|
> ||P.I.|0.259|0.924|2.813|3.071|1.229|0.204|
>
> **Answer to Q2:**
>
> Thank you for your question. The graphon model offers a wide range of useful applications in the analysis of network data. Below, we summarize the key uses of the model and its potential downstream tasks:
>
> 1) *Network structure visualization and interpretation.*   The graphon provides a interpretable representation of the underlying structure within a network. This allows researchers to gain insights into the composition of the network, including identifying key motifs or structural patterns, which is often challenging with raw connection data alone [1].
>
> 2) *Network simulation and scalability.* The graphon can be used as a generative tool for simulating networks. By sampling from the graphon model, networks of arbitrary size can be generated. This is especially useful in domains where full data acquisition is costly or impractical, allowing us to simulate large-scale networks without the need for complete data [2].
>
> 3) *Handling missing data and edge prediction*. The graphon model is flexible in handling missing data. Graphons are naturally suited for edge prediction tasks in incomplete or partially observed networks, making them an effective tool for tasks such as link prediction and network completion.
>
> 4) *Network comparison.*  Since the graphon can be interpreted as a density or intensity function over the network, it is particularly useful for comparing different networks. When only the connection probability matrices $P_1, P_2$ are available, meaningful comparison becomes difficult, especially if the networks differ in size, because the matrices are not directly comparable in such cases. Graphon estimation, however, is independent of network size, making it a more flexible tool. For instance, suppose we obtain two estimated graphons, $\hat{f}_1$ and $\hat{f}_2$. We can then use a statistic $\sup _{x,y} |\hat{f} _1(x,y) - \hat{f} _2(x,y)|$ to perform hypothesis testing on whether the two networks are generated from the same underlying mechanism. If this supremum difference is sufficiently large, we can conclude that the networks have different generative structures.
>
> 5) *Motif prediction.* The expected density of a fixed motif $F$ (e.g., a triangle or star) in a large random graph generated from $f$ is given by its homomorphism density: $t(F,f) = \\int_{[0,1]^k} \\prod_{(i,j) \\in E(F)} f(x_i, x_j) \, dx_1 \\cdots dx_k,$ where $k = |V(F)|$. Additionally, other important graph features can be expressed as functions of the graphon. For instance, the expected transitivity is given by $C = \frac{\int_{[0,1]^3} f(x,y) f(y,z) f(z,x) \, dx \, dy \, dz}{\int_{[0,1]^3} f(x,y) f(y,z) \, dx \, dy \, dz}.$  Graphon estimation enables the direct prediction of higher-order features by plugging in the estimated graphon. Moreover, this procedure requires only a constant number of matrix operations per sample. In contrast, counting motifs explicitly is computationally expensive (see [3] for details).

---

> ### Author Response · Authors · 2025-08-02
>
> **Answer to Q3:**
>
> Thank you for your valuable suggestion. We have conducted additional experiments for the scenario where the true rank is 2 and the estimated rank is 1. The results is presented in Table R5, and will be included in the revised version of the paper.
>
> |ID|MSE($\times10^{-4}$)|Std.dev of MSE($\times10^{-6}$)|Max.error($\times10^{-2}$)|Std.dev of max.error($\times10^{-3}$)|Runtime(s)|Std.dev of runtime(s)|
> |--|--------------------|--------------------------------|--------------------------|-------------------------------------|----------|----------------------|
> |4|15.559|5.939|30.818|17.708|0.205|0.017|
> |5|1.810|0.620|10.320|8.962|0.365|0.048|
> |6|71.987|29.280|15.086|2.447|0.331|0.069|
>
> As shown in the table, the estimation error increases significantly when rank-2 graphons are incorrectly selected as rank 1, since an essential component is omitted during estimation. However, our rank selection experiments indicate that while the selected rank may sometimes be higher than the true rank, we seldom select a rank lower than the actual one, which would otherwise lead to large errors. This demonstrates the robustness of our method in practice.
>
> [1] Sischka B. Graphon models for network data: estimation, extensions and applications[D]. lmu, 2023.
>
> [2] Gao S, Caines P E. Graphon control of large-scale networks of linear systems[J]. IEEE Transactions on Automatic Control, 2019, 65(10): 4090-4105.
>
> [3] Jin J, Ke T, Sui B, et al. Counting Cycles with Deepseek[J]. arXiv preprint arXiv:2505.17964, 2025.

---

> ### Comment · Reviewer_n6Ak · 2025-08-05
> **Thank you for the further responses.**
>
> While a concern regarding readability, presentation, and reproducibility still remains, the issues raised above have been adequately addressed. Therefore, I am happy to adjust my score accordingly.

---

### Official Review · Reviewer_1DRW · 2025-07-03

**Clarity:** 2
**Significance:** 3
**Originality:** 3
**Rating:** 5
**Confidence:** 5

**Summary:**

The author(s) propose a low-rank graphon modelling for networks. The estimation approach reconcile the estimation of the graphon function and the corresponding connection probability matrix. The author(s) propose a sequential efficient algorithm baed on subgraph counts and interpolation. They also prove theoretical consistency results in sup-norm for the connection matrix and the graphon function estimation. The proposed approach is validated on simulated data examples and on real world network data.

**Questions:**

I think the low-rank modelling approach using graphon models for networks is a novel formulation indeed. Especially, the contribution in the direction of joint estimation of connection matrix and the graphon function itself is a long-standing problem in the literature of networks. So indeed this is valuable.

I have a couple of minor queries for the author(s)

-I may have missed this point but how does one ensure the uniqueness of the estimated graphon function $\hat{f}^{+}(u,v)$ ?
-What happens if there are ties in the degree sequence $\lbrace d_i\rbrace_{i=1}^n$? How do you do the sorting then i.e. how breaking of ties is implemented?
-Does the additional assumption 3.7 means in certain cases one can only estimate $P$ and not the graphon function $f$?
-Why ratio is being used in (A.5) to select unknown rank $r$ instead of the difference $|\lambda_{k+1}-\lambda_k|$?
-Theorem 3.8 says the sup-norm rate is independent of $r$ but practically speaking, not all networks can have same rate of estimation since the rank may vary. Am I missing something here?

**Ethical Concerns:**

["NO or VERY MINOR ethics concerns only"]

**Final Justification:**

The author(s) put in a sincere and detailed effort in answering queries posed by myself and other reviwers. I am happy with the author(s)'s response and would like to maintain my score "Accept".

**Limitations:**

yes

**Quality:**

3

**Strengths And Weaknesses:**

strengths:

1. A novel modelling approach of network data via low-rank graphon modelling
2. Estimating jointly the graphon function and the connection matrix via an efficient algorithm based on subgraph counts and interpolation
3. Strong theoretical results on sup-norm consistency of graphon function estimation and connection matrix.
4. Scalable algorithm that has favourable time complexity over its neighbours.
5. The assumptions in the theoretical framework often matches with those in SBM or RDPG network models.

weakness

1. For clarity and understanding, the section on low-rank modelling with rank $r > 2$ should be rewritten especially the graphon function estimation. The subgraph count based idea along with sorting may facilitate reader's understanding with a toy example.
2. Some motivation/discussion was lacking why graphon function estimation would be useful beyond estimation the connection matrix.
3. As it seems, the motivational example of $r=1$ uses the method of moments approach in estimating the $p_{ij}$. Some discussion on that would have been good as method of moments has been used in network parameter estimation earlier in the literature as well.

---

> ### Author Rebuttal · Authors · 2025-07-31
>
> Thank you for your positive feedback on our paper. We appreciate your recognition of the novelty of our approach, theory, and the effectiveness of the algorithms. In the following, we will address each concern that you raised one by one.
>
> **1. Regarding the clarity of the section on low-rank modeling with rank $r > 2$.**
>
> Thank you for this helpful suggestion. We agree that the discussion of low-rank modeling for $r > 2$, particularly in the context of graphon estimation, could benefit from additional clarity. In the revised version, we will rewrite this section, as suggested, to better convey the key ideas and provide more intuitive explanations. To aid understanding, we will include a toy example based on the graphon $f(u,v) = \cos(u - v)$. While omitted here due to space constraints, we will incorporate it in the revision to make the presentation more accessible.
>
> **2. Regarding why graphon function estimation is useful beyond estimating the connection matrix.**
>
> Thank you for this insightful question. We clarify that graphon estimation provides a richer and more general description of network structure than the connection probability matrix $P$. While $P$ captures marginal probabilities for edges in a specific network instance, it lacks information about how edges jointly interact. In contrast, the graphon $f$ defines a generative model over a family of networks and encodes global structural properties, including the distribution of motifs and other higher-order patterns.
>
> For example, the expected density of a fixed motif $F$ (e.g., a triangle or star) in a large random graph generated from $f$ is given by its homomorphism density: $t(F,f) = \\int_{[0,1]^k} \\prod_{(i,j) \\in E(F)} f(x_i, x_j) \, dx_1 \\cdots dx_k,$ where $k = |V(F)|$. For triangles, this quantity captures the limiting probability that three nodes form a triangle, a fundamentally joint property that $P$ alone cannot characterize. Similarly, the expected transitivity can be expressed as $C = \\frac{\\int_{[0,1]^3} f(x,y) f(y,z)f(z,x) dxdydz}{\\int_{[0,1]^3} f(x,y) f(y,z)  dxdydz}.$
>
> Graphon estimation thus enables direct prediction of such higher-order features. It also facilitates comparison between networks of different sizes by analyzing their underlying generative mechanisms, independently of the specific realizations captured in $P$.
>
> To support this, we conducted an experiment using graphons with IDs 4-6. We generated networks with 2000 nodes, randomly removed 10\% of the nodes, estimated the graphon from the subgraph, and regenerated networks from the estimate. The mean absolute errors in triangle counts and transitivity (normalized appropriately) over 100 trials are shown below in Table R1.
>
> *Table R1: Triangle and transitivity errors across 100 repetitions.*
>
> | Graphon ID | Triangle Error $(\times 10^{-4})$ Mean | Triangle Error $(\times 10^{-4})$ Std. Dev. | Transitivity Error $(\times 10^{-3})$ Mean | Transitivity Error $(\times 10^{-3})$ Std. Dev. |
> |------------|---------------------------------------|--------------------------------------------|--------------------------------------------|------------------------------------------------|
> | 4          | 2.942                                 | 5.309                                      | 0.696                                      | 6.240                                          |
> | 5          | 1.863                                 | 2.107                                      | 1.506                                      | 1.749                                          |
> | 6          | 0.914                                 | 1.192                                      | 1.346                                      | 0.453                                          |
>
> These consistently low errors demonstrate that key higher-order features are well preserved, even with subsampling, highlighting the structural fidelity and generalizability of graphon estimation. We will incorporate this discussion and table in the revised paper.
>
> **3. Regarding discussion of the method of moments (MoM) used in network parameter estimation.**
>
> Thank you for your question. The method of moments (MoM) has been employed in the network literature, including [3], where MoM was used to estimate parameters based on subgraph counts, and [4], where the authors analyzed the finite-sample distribution of MoM estimators for network motifs via Edgeworth expansions.
>
> In our work, we align network sparsity using MoM when $r = 1$, which is inspired by [3]. We will include explicit citations and discussion of these relevant works in the revised version.
>
> **4. Regarding the uniqueness of the estimated graphon function $\\hat{f}^{+}(u,v)$ and the ties in the degree sequence $\\{d_i\\}$**
>
> Thank you for your question. If there are ties in the degree sequence - for example, if $d_1 = d_2$ and they represent the smallest degrees - then for $\\frac{1}{n+1} \\le v < \\frac{2}{n+1}$, $h(v)$ will always be equal to $d_1$ (which is also equal to $d_2$). In fact, ties do not affect the function $h(v)$: the degrees with ties can be ordered in any sequence, and the resulting $h(v)$ will remain unchanged. This further ensures the uniqueness of $\\hat{f}^{+}(u,v)$.
>
> **5. Regarding Assumption 3.7.**
>
> Thank you for your question. The short answer is yes. Assumption 3.7 provides an additional condition for estimating the underlying graphon function $f$ (beyond the connection probability matrix $P$). Without such conditions, the graphon is non-identifiable--it is only defined up to measure-preserving transformations. It is well-known that in this case, we can typically only estimate the probability matrix $P$ (see, for example, Section 10.2 in [2] for discussions on identifiability). Therefore, when estimating the graphon itself, the literature generally requires additional conditions (such as [9]). Assumption 3.7 plays a similar role, enabling the recovery of both the connection probability matrix $P$ and the graphon $f$.
>
> **6. Regarding why a ratio is used in selecting the unknown rank $r$.**
>
> Thank you for your question. The eigenratio method is a well-established approach in the statistics literature (e.g., [5], [6]). It offers robustness to scaling and noise, especially in settings where the eigenvalues decay gradually or have non-uniform magnitudes (see also [7]).
>
> In our context, the use of the ratio helps highlight significant drops in the eigenvalue sequence while mitigating the influence of slow decay or scale variability. Algorithm 4 selects the correct rank $r$ with high probability as the threshold $\\tau$ asymptotically approaches zero at a certain rate. We will clarify this connection and cite the relevant literature in the revision.
>
> **7. Regarding the relation between the sup-norm rate and the rank $r$.**
>
> Thank you for your question. In this paper, we treat $r$ as fixed, which is why it does not explicitly appear in the stated error rate--it is absorbed into the constant factor. In practice, $r$ is typically unknown and may vary, which motivates the inclusion of Algorithm 4 to estimate it.
>
> We agree that the error rate should reflect the dependence on $r$ more explicitly. Since the rate deteriorates as $r$ increases, we will include this dependence in the revised statement of the error rate. The case where $r$ grows with $n$ is an interesting and important direction, which we leave for future work.
>
> [1] Borgs, Christian, Jennifer Chayes, László Lovász, Vera T. Sós, Balázs Szegedy, and Katalin Vesztergombi. "Graph limits and parameter testing." In Proceedings of the thirty-eighth annual ACM symposium on Theory of computing, pp. 261-270. 2006.
>
> [2] Lovász, László. Large networks and graph limits. Vol. 60. American Mathematical Soc., 2012.
>
> [3] Bickel, Peter J., Aiyou Chen, and Elizaveta Levina. "The method of moments and degree distributions for network models." (2011): 2280-2301.
>
> [4] Zhang, Yuan, and Dong Xia. "Edgeworth expansions for network moments." The Annals of Statistics 50, no. 2 (2022): 726-753.
>
> [5] Lam, C., \& Yao, Q. (2012). Factor modeling for high-dimensional time series: inference for the number of factors. The Annals of Statistics, 694-726.
>
> [6] Ahn, S. C., \& Horenstein, A. R. (2013). Eigenvalue ratio test for the number of factors. Econometrica, 81(3), 1203-1227.
>
> [7] Cai, T. Tony, Dong Xia, and Mengyue Zha. "Optimal differentially private PCA and estimation for spiked covariance matrices." arXiv preprint arXiv:2401.03820 (2024).
>
> [8] Von Luxburg, Ulrike. "A tutorial on spectral clustering." Statistics and computing 17, no. 4 (2007): 395-416.
>
> [9] Chan, Stanley, and Edoardo Airoldi. "A consistent histogram estimator for exchangeable graph models." In International Conference on Machine Learning, pp. 208-216. PMLR, 2014.

---

> > ### Comment · Reviewer_1DRW · 2025-08-04
> > **Response to author rebuttal**
> >
> > I thank the author(s) for their detailed response to my queries. I appreciate the table R1 which shows preservation of the higher order properties for subgraph. I am happy with the author(s)'s response and would like to maintain my score "Accept".

---

### Comment · Area_Chair_nAtW · 2025-08-01
**The time to start author-reviewer discussions**

Dear all reviewers,

The author rebuttal period has now concluded, and authors' responses are
available for the papers you are reviewing. The Author-Reviewer Discussion
Period has started, and runs until August 6th AoE.

Your active participation during this phase is crucial for a fair and
comprehensive evaluation. Please take the time to:

- Carefully read the author responses and all other reviews.
- Engage in a constructive dialogue with the authors, clarifying points,
  addressing misunderstandings, and discussing any points of disagreement.
- Prioritize responses to questions specifically addressed to you by the authors.
- Post your initial responses as early as possible within this window to
  allow for meaningful back-and-forth discussion.

Your insights during this discussion phase are invaluable.
Thank you for your continued commitment to the NeurIPS review process.

Bests,
Your AC

---

### Decision · Program_Chairs · 2025-09-17

**Decision:**

Accept (poster)

**Comment:**

This paper proposes a novel and computationally efficient method for
estimating low-rank graphons, a significant problem in network analysis.
The core contribution is an approach that jointly estimates the connection
probability matrix and the underlying graphon function using subgraph
counts, supported by strong theoretical guarantees on the estimation error.

Initially, this was a classic borderline paper. Reviewers unanimously
praised the technical novelty and theoretical rigor but raised significant
and valid concerns about the empirical validation and presentation clarity.
The original submission lacked quantitative evaluation on real-world data,
provided limited simulation results, and did not sufficiently motivate the
practical utility of graphon estimation beyond recovering the connection
matrix.

However, the authors provided an exceptionally thorough and convincing
rebuttal that fundamentally strengthened the submission. They conducted
numerous new experiments, including a quantitative evaluation of motif
prediction on real-world data, additional metrics for graphon estimation
error, and new analyses under varying sparsity and rank-mismatch
conditions. This extensive new evidence successfully addressed the primary
concerns of the reviewers, leading two of them to raise their scores in
favor of acceptance. It needs to be emphasised that the current state of
this submission (specifically, the section of numerical experiments) has
not met the standard of NeurIPS publication, and these experiment
supplements in the rebuttal must be included to the final preparation.

While some minor concerns remain regarding the presentation and certain
theoretical assumptions, the consensus is that the paper's novel
methodology, strong theoretical guarantees, and now substantially improved
empirical support make it a valuable contribution. The paper is recommended
for acceptance.